# Aborting meiosis allows recombination in sterile diploid yeast hybrids

Simone Mozzachiodi [1,2,10], Lorenzo Tattini [1,10], Agnes Llored[1], Agurtzane Irizar[1], Neža Škofljanc[1],
Melania D'Angiolo[1], Matteo De Chiara[1], Benjamin P. Barré[1], Jia-Xing Yue [1,7], Angela Lutazi[1], Sophie Loeillet [3],
Raphaelle Laureau[3,8], Souhir Marsit [3,4,9], Simon Stenberg [5], Benoit Albaud[6], Karl Persson [5],
Jean-Luc Legras [4], Sylvie Dequin[4], Jonas Warringer [5], Alain Nicolas[1,2,3] & Gianni Liti [1✉]

Hybrids between diverged lineages contain novel genetic combinations but an impaired meiosis often makes them evolutionary dead ends. Here, we explore to what extent an aborted meiosis followed by a return-to-growth (RTG) promotes recombination across a panel of 20 *Saccharomyces cerevisiae* and *S. paradoxus* diploid hybrids with different genomic structures and levels of sterility. Genome analyses of 275 clones reveal that RTG promotes recombination and generates extensive regions of loss-of-heterozygosity in sterile hybrids with either a defective meiosis or a heavily rearranged karyotype, whereas RTG recombination is reduced by high sequence divergence between parental subgenomes. The RTG recombination preferentially arises in regions with low local heterozygosity and near meiotic recombination hotspots. The loss-of-heterozygosity has a profound impact on sexual and asexual fitness, and enables genetic mapping of phenotypic differences in sterile lineages where linkage analysis would fail. We propose that RTG gives sterile yeast hybrids access to a natural route for genome recombination and adaptation.

[1] Université Côte d'Azur, CNRS, INSERM, IRCAN, Nice, France. [2] Meiogenix, 38, rue Servan, Paris 75011, France. [3] Institut Curie, Centre de Recherche, CNRS-UMR3244, PSL Research University, Paris 75005, France. [4] SPO, Université Montpellier, INRAE, Montpellier SupAgro, Montpellier, France. [5] Department of Chemistry and Molecular Biology, University of Gothenburg, Gothenburg, Sweden. [6] Institut Curie, ICGEX NGS Platform, Paris 75005, France. [7] Present address: State Key Laboratory of Oncology in South China, Collaborative Innovation Center for Cancer Medicine, Sun Yat-sen University Cancer Center, Guangzhou, China. [8] Present address: Department of Genetics and Development, Hammer Health Sciences Center, Columbia University Medical Center, New York, NY, USA. [9] Present address: Département de Biologie Chimie et Géographie, Université du Québec à Rimouski, Rimouski, Québec, Canada. [10] These authors contributed equally: Simone Mozzachiodi, Lorenzo Tattini. ✉email: gianni.liti@unice.fr

**M**eiotic recombination is a primary source of genetic diversity in species undergoing sexual reproduction and is the main feature of sex conserved across eukaryotic kingdoms[1]. During meiosis, crossovers enable haploid gametes to receive one copy of each chromosome and thereby prevent genetic imbalance in the offspring that often decreases fitness[2]. However, reproductive barriers acting before (pre-zygotic) or after (post-zygotic) zygote formation[3] can arise during species evolution, with post-zygotic barriers leading to hybrid sterility[4] across the tree of life[5].

Hybrids within the *Saccharomyces* genus have served as models to elucidate the mechanisms contributing to post-zygotic reproductive isolation[6]. Sterile *Saccharomyces* intraspecies and interspecies hybrids have been repeatedly isolated in both wild and domesticated environments[7]. Relaxed selection on sexual reproduction in domesticated populations of *S. cerevisiae*, including intraspecies hybrids, has led to the accumulation of loss-of-function mutations in genes involved in gametogenesis, i.e., "sporulation" in yeast biology, and to severe sterility[8]. Chromosomal rearrangements between the subgenomes of a hybrid can also lead to sterility owing to aberrant chromosome pairing and segregation[9–11]. The Malaysian *S. cerevisiae* lineage represents the most dramatic karyotype-driven speciation example as it contains 5 rearranged chromosomes that isolate it reproductively from other *S. cerevisiae* lineages, despite retaining high levels of sequence similarity to these[12,13]. In contrast, *Saccharomyces* interspecies hybrids produce inviable gametes because of the large DNA sequence divergence between parental subgenomes, namely heterozygosity, which suppresses recombination and leads to chromosome missegregation[14]. Therefore, many *Saccharomyces* intraspecies and interspecies hybrids have only very limited possibilities to evolve through meiosis and some are completely sterile. Moreover, wild yeasts reproduce sexually approximately only once every 1000 asexual generations, thus further limiting the role of meiosis in both intraspecies and interspecies evolution[15].

We recently showed that aborting meiosis in a fertile *S. cerevisiae* laboratory hybrid and returning the cells to mitotic growth, a process known as "return-to-growth" (RTG)[16], reshuffles the parental diploid genome and produces extensive regions of loss-of-heterozygosity (LOH)[17]. LOHs resulting from single crossing-over generate terminal events that extend to the chromosome ends, whereas interstitial LOHs in chromosome cores can be ascribed to gene conversions or double crossovers. In *S. cerevisiae*, meiosis is initiated by starvation, and chromosomes are replicated during the meiotic S-phase (Fig. 1a). Next, as in most eukaryotes[18], a homolog of an archaea DNA topoisomerase VI, Spo11p in yeast, generates genome-wide double-strand breaks (DSBs)[19]. The repair of these DSBs produces joint DNA molecules and ultimately leads to recombination through chromosomal crossover (CO) and non-crossover (NCO) events. Yeast cells that are brought back to a nutrient-rich environment before the commitment to complete meiosis abort the meiotic programme and express genes promoting mitotic division, thus entering the RTG process[20,21]. RTG cells repair the DSBs, bud-like mitotic cells and segregate their recombined chromatids with no further DNA replication[22]. This process generates a mother and a daughter cell that both maintain the parental diploid state, as they do not complete the two chromosomal segregation events that occur in complete meiosis, but contain different rearranged genotypes. Here, we show that RTG allows recombination in *Saccharomyces* hybrids that are sterile due to common reproductive barriers, and we propose RTG as a powerful alternative to completed meiosis that may play an unanticipated role in the evolution of yeast hybrid genomes in nature.

## Results

**An incomplete meiosis supports recombination in a sterile intraspecies hybrid.** Deleterious variants in meiotic genes can lead to various meiotic defects and sterility. However, mutations that impair middle or late meiotic progression should not prevent RTG, if cells can resume mitotic-like chromosome segregation, and budding[22,23] (Fig. 1a). Therefore, we probed whether RTG allows *S. cerevisiae* (Sc) to the bypass mutational barriers in meiosis by deleting *NDT80*, the master regulator of middle and late meiotic genes, in an otherwise fertile ScS288C/ScSK1 intraspecies lab-hybrid. First, we confirmed by DAPI staining that *ndt80Δ* cells were all arrested before the first meiotic division (MI) after 8 h of sporulation induction, whereas ≥50% of the wild-type ScS288C/ScSK1 cells had already completed MI (≥ 2 nuclei) at this time-point. Then, we evolved these hybrid cells through one cycle of RTG and sequenced the genome of 12 mother–daughter *ndt80Δ* RTG pairs (*n* = 24) isolated 8 h after sporulation induction. As a comparison, we re-analysed the mother–daughter RTG genomes (*n* = 22) of wild-type ScS288C/ScSK1 hybrid cells returned to growth 4 and 5 h after sporulation induction. At these time points, wild-type cells experience different levels of DSBs but are not yet committed to complete meiosis[17]. We analysed short-read sequencing data using an integrated framework that enables a high-resolution view of LOHs, even if supported by only a single marker[24]. All the *ndt80Δ* clones had highly recombined diploid genomes (Fig. 1b) with an average of 88 LOHs per clone, summing terminal ($n_{total}$ = 181) and interstitial ($n_{total}$ = 1933) LOHs. The number of interstitial LOHs was higher in *ndt80Δ* than in wild-type RTG clones (Fig. 1c, Supplementary Fig. 1a), and the former also had more markers lying in the region of non-reciprocal recombination between the mother–daughter RTG pairs (*p* value = 3.663 × $10^{-06}$, Welch's *t* test) (Supplementary Data Set 13). Consistently, the *ndt80Δ* RTG clones accumulated more NCOs per clone[25]. In contrast, large LOHs (>10 kbp) were rarer in *ndt80Δ* than in wild-type RTG clones (Supplementary Figs. 1a, 2a). Despite these differences in LOH size, *ndt80Δ* and wild-type RTG clones showed a similar median fraction of markers in LOH (Supplementary Data Set 6), but the distribution was more homogeneous in the *ndt80Δ* RTG cells compared to the wild-type (Fig. 1d). This finding showed that *ndt80Δ* cells had uniformly progressed up to prophase I when meiosis was aborted, whereas in wild-type cells meiosis was arrested at different stages of the early meiotic phase. The frequency of LOH events increased from the centromere towards chromosome ends, in accordance with centromeres being cold meiotic recombination regions (Supplementary Fig. 1c). Finally, we found no bias in LOH formation towards either of the two parental subgenomes in the data sets, in line with RTG generating complementary recombined genomes with few non-reciprocal events[17] (Fig. 1e, Supplementary Data Set 13). Overall, we conclude that RTG allows a hybrid that is unable to complete meiosis owing to the lack of a functional key meiotic gene to generate highly recombined diploid genomes.

**Profiling RTG efficiency across hybrid diversity.** Next, we asked to what extent RTG promotes recombination in hybrids whose variable degrees of sterility derive either from sequence divergence (heterozygosity) or structural differences between the subgenomes. To answer this question, we exploited natural variation to generate a panel of 19 diploid genetic backgrounds, comprising four fully homozygous *S. cerevisiae* (Sc) strains, 7 *S. cerevisiae* intraspecies hybrids (Sc/Sc) and eight interspecies hybrids between *S. cerevisiae* and its sister species *S. paradoxus* (Sp/Sc) (Fig. 2a–b, Supplementary Data Set 1). Then, we developed a simple genetic system

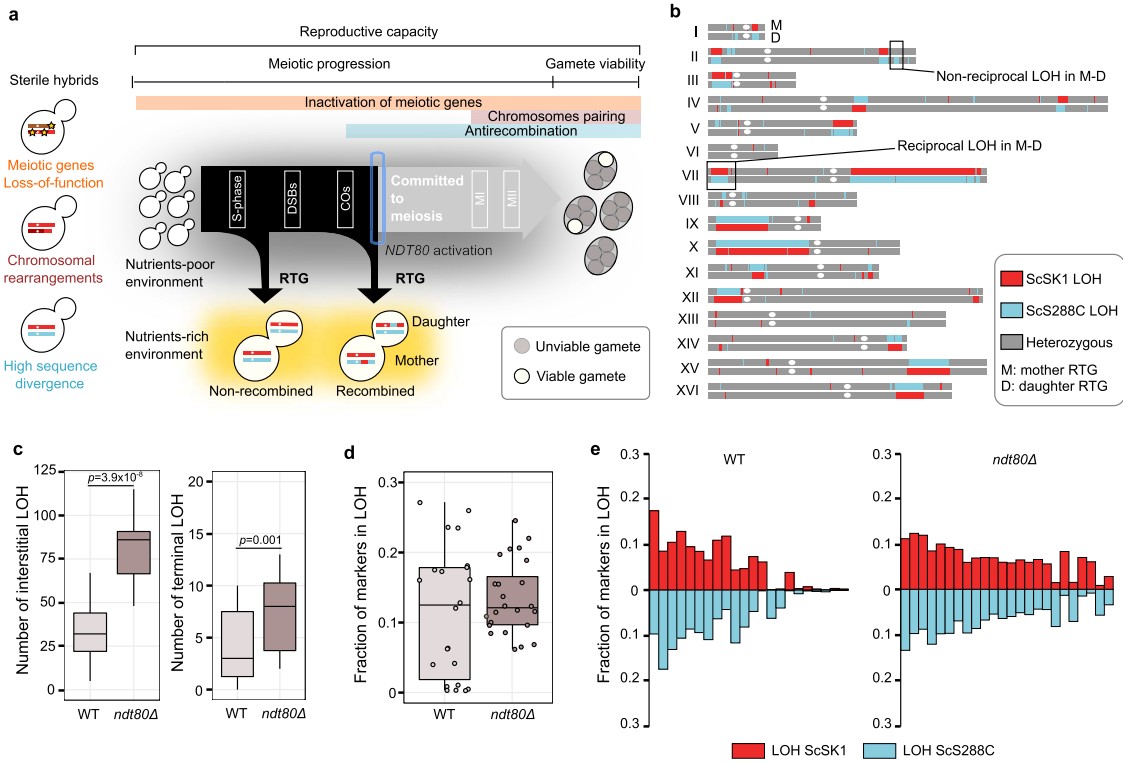

**Fig. 1 RTG paradigm and genomic landscape of ndt80Δ evolved RTG clones. a** Inactivation of essential meiotic genes (yellow stars), genomic rearrangements and high levels of heterozygosity in hybrid genomes can impair meiotic progression or reduce gamete viability. RTG represents an alternative route to hybrid evolution that creates LOH regions. **b** LOH map in a mother/daughter (top/bottom, respectively) RTG pair derived from the *ndt80Δ* hybrid. LOH blocks can be tracked by genotyping single-nucleotide markers which give a homozygote readout. **c** Boxplots of the number of interstitial (left) and terminal (right) LOH events in the wild-type (WT, $n = 22$) and *ndt80Δ* ($n = 24$) RTG clones derived from ScS288C/ScSK1. The *ndt80Δ* RTGs have more interstitial ($p$ value $= 3.9 \times 10^{-8}$) and terminal ($p$ value$=0.001$) LOHs. Both tests are one-tailed Wilcoxon rank-sum tests with continuity correction. Testing only the mother cells in both data sets ($n = 12$ *ndt80Δ*, $n = 11$ WT) confirmed that *ndt80Δ* mothers accumulated more interstitial ($p$ value $= 5.033 \times 10^{-8}$) and terminal ($p$ value $=0.0053$) LOHs, using one-tailed Wilcoxon rank-sum test with continuity correction. No parental bias was detected by comparing the average number of LOHs towards one of the two subgenomes in WT (Welch's $t$ test two-tailed, $p$ value $= 0.67$) and *ndt80Δ* data sets (Welch's $t$ test two-tailed, $p$ value $= 0.25$). **d** Boxplot reporting the fraction of markers in LOH. Each point represents an RTG sample. The distribution of the WT samples ($n = 22$) is broader and shows a larger interquartile range compared with the *ndt80Δ* data set ($n = 24$). **e** Fraction of homozygous markers for both parental alleles for the WT and *ndt80Δ* data sets. Each bar represents a sequenced clone. The *ndt80Δ* RTGs show a lower variation compared with the wild-type, both considering a pooled data set ($n = 24$) (Fligner-Killeen test two-tailed, $p$ value $= 0.0008$) and considering only the mothers from each data sets (Fligner-Killeen test two-tailed, $p$ value $= 0.02$). The boxplot is defined as follows: the box is delimited by the first quartile (Q1) and the third quartile (Q3). The line that separates the box is the median. Whiskers are defined as: upper whisker $= \min(\max(x), Q3 + 1.5 \times IQR)$; lower whisker $= \max(\min(x), Q1 - 1.5 \times IQR)$, where: $x$ is the data, Q1 is the first quartile, Q3 is the third quartile and IQR is the interquartile range ($IQR = Q3 - Q1$).

to measure RTG-induced LOH rates (Supplementary notes) at the *LYS2* locus on chromosome II (Fig. 2c). We replaced one of the *LYS2* alleles with a *URA3* gene and measured how often this *URA3* marker was lost due to LOH in cells returned to growth after 6 h of meiosis induction (T6), before the commitment to the meiosis of the fastest sporulating strain (Supplementary Fig. 3a), and control cells (T0). We calculated the LOH rates at T6 and T0 ($R_{T6}$ and $R_{T0}$, respectively), and we derived the efficiency of RTG-induced LOH using two metrics: the "LOH ratio" ($R_{T6}/R_{T0}$) and the "LOH difference" ($R_{T6} - R_{T0}$) (Fig. 2d and Supplementary Data Set 4). We observed pronounced genetic background effects with fast-sporulating strains promoting efficient recombination upon RTG (Fig. 2e, Supplementary Data Set 2). Among the homozygous diploids, the ScNA/ScNA showed both a high LOH ratio (~21) and the largest LOH difference (Fig. 2d), consistently with its faster meiotic progression. On the contrary, the other three homozygous diploids had worse sporulation efficiency and synchrony (Fig. 2d and Supplementary Fig. 3b), preventing the

production of a significant number of recombinant RTGs. All the intraspecies hybrids with a ScNA subgenome also showed a significant increase of LOH rates upon RTG, with the LOH ratio ranging from 3 to 42, whereas other intraspecies hybrids showed no significant increase (Fig. 2d, Supplementary Data Set 4). We detected a 23-fold increase in LOH upon RTG in the ScMA/ScNA hybrid, whose Malaysian subgenome is highly non-collinear compared to other *S. cerevisiae* yeasts. Thus, the non-collinearity between several homologous chromosomes (other than chromosome II) did not reduce RTG-induced LOH at the *LYS2* locus. The LOH difference in the ScMA/ScNA hybrid was also comparable with the values measured for the other intraspecies hybrids with collinear subgenomes (e.g., ScNA/ScWA or ScWE/ScNA). In contrast, all the interspecies *S. cerevisiae*/*S. paradoxus* hybrids had a lower basal LOH rate ($R_{T0}$) during mitosis (10-fold on average) and RTG enhanced LOH less than in intraspecies hybrids both in terms of LOH ratio and LOH difference (Fig. 2d, Supplementary Data Set 4). Nevertheless, all of the three interspecies hybrids with

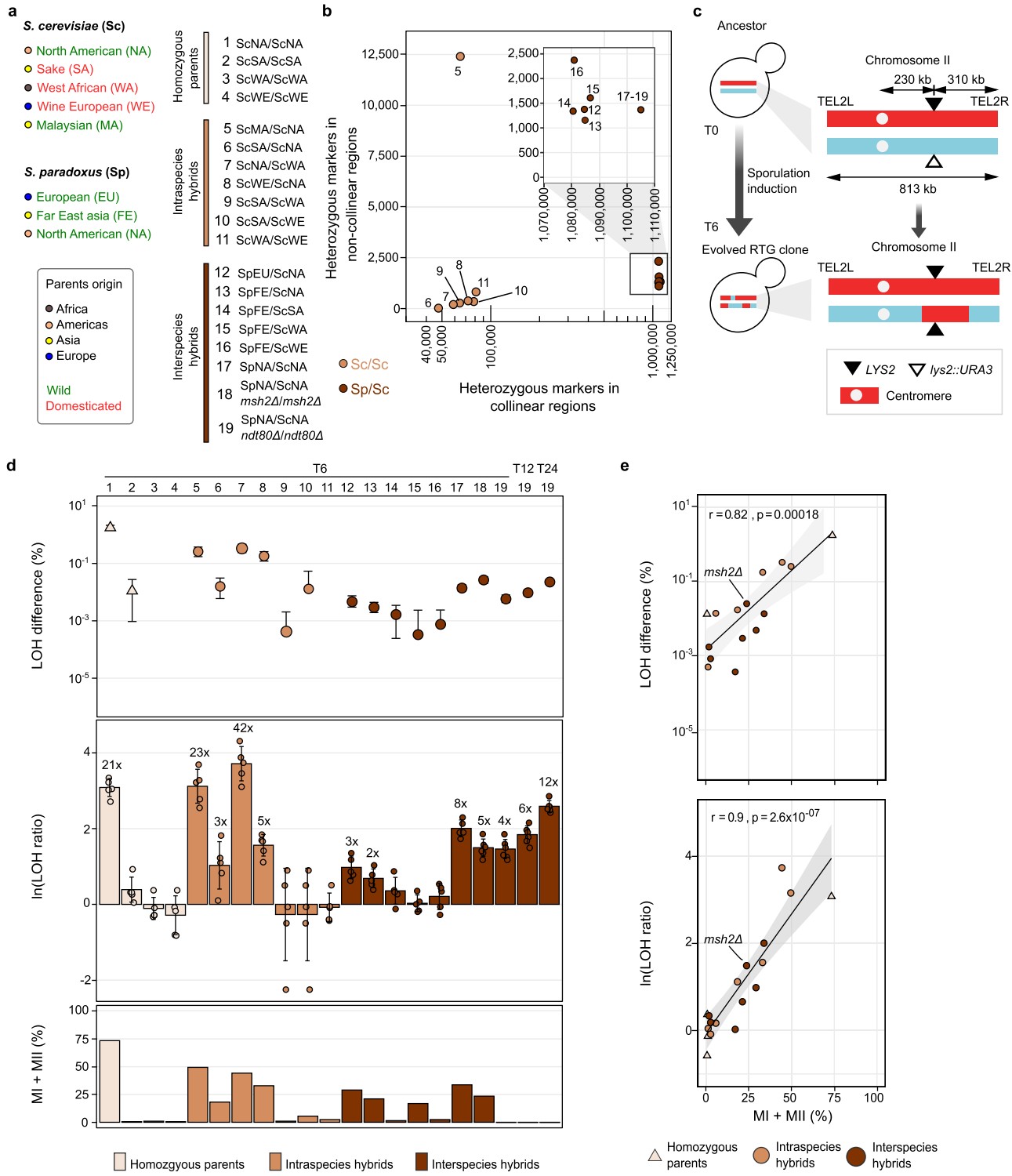

a fast-sporulating (Supplementary Fig. 3) ScNA subgenome showed significant LOH rate increases (Supplementary Data Set 4) as well as higher LOH ratios and differences compared with other interspecies hybrids (Fig. 2d). The lower LOH rates of the interspecies hybrids compared to the intraspecies hybrids at both T0 and T6 suggest that high heterozygosity strongly inhibits recombination in interspecies hybrids, through the anti-recombination mechanism, as it does during meiosis[26]. To test this scenario, we deleted the *MSH2* gene, which encodes a key protein of the mismatch-binding machinery, in both subgenomes of the SpNA/

ScNA interspecies hybrid. Indeed, we observed increased LOH rates in both T0 (3.6-fold) and T6 (twofold) samples compared with the wild-type SpNA/ScNA hybrid (Supplementary Data Set 4), despite meiotic progression being slower in the *msh2Δ* mutant (Fig. 2d and Supplementary Fig. 3b). This is consistent with the mismatch-repair machinery acting to prevent RTG recombination in hybrids with highly diverged subgenomes.

Finally, we generated a SpNA/ScNA *ndt80Δ* with both high heterozygosity and meiotic progression defects, and we measured LOH ratios and differences after 6 (T6), 12 (T12) and 24 (T24)

**Fig. 2 Quantifying RTG-induced recombination across hybrid diversity. a** Geographical (coloured circle) and ecological (coloured name) origins of the parental strains (left) used for generating the diploid backgrounds (right). Diploids are grouped according to their level of heterozygosity as: homozygous parents (light brown), intraspecies hybrids (brown) and interspecies hybrids (dark brown) and the same colour codes apply to **b**, **d** and **e**. **b** Level of heterozygosity across the hybrid panel with the number of markers detected in non-collinear and collinear regions. Each data point is labelled with a number/colour encoded according to **a**. The four homozygous parents are not reported. **c** URA3-loss assay used for measuring RTG-induced recombination rates. **d** Top: the y axis reports in logarithmic scale the percentage of cells growing in 5-FOA measured with the URA3-loss assay and quantified with the LOH difference ($R_{T6}-R_{T0}$) where $R_{T6}$ and $R_{T0}$ are the LOH rates at T6 and T0, respectively (Supplementary notes). Error bars represent standard deviations. Each point represents the average of $n = 5$ independent replicates. Samples having a negative LOH difference (non-significant differences between control and RTG sample) are not reported. Middle: cell growth quantified with the natural logarithm of the LOH ratio ($R_{T6}/R_{T0}$). Error bars represent standard deviations. Each point represents the average of $n = 5$ independent replicates. The number on the top of each bar indicates the linear fold increase. Bottom: meiotic progression after 12 h measured as the percentage of cells that passed the first (MI) and the second (MII) meiotic divisions (quantified with fluorescence microscopy of DAPI-stained cells). **e** Correlation between the meiotic progression (MI + MII cells after 12 h in sporulation medium) and the LOH difference (top, logarithmic scale) as well as the natural logarithm of the LOH ratio (bottom). r is the Pearson's correlation coefficient, the p value of the correlation is calculated as a two-sided test. In both plots, the grey area represents the 95% confidence interval. Samples showing negative LOH differences were removed to avoid a bias towards correlation.

hours of sporulation induction. We observed that RTG recombination increased with the time spent in gametogenesis, in accordance with the fraction of cells engaged in RTG increasing with time (Fig. 2d). Hence, genetically similar cells in clonal populations are nevertheless quite heterogeneous in their meiotic progression, with only a minor fraction of cells having committed to recombination at 6 h. Thus, the absence of RTG-induced recombination in a hybrid may reflect that cells had not progressed sufficiently in their meiosis to engage in recombination and to produce recombinant RTGs.

Our results indicate that both the meiotic progression and the level of sequence divergence affect the LOH metrics measured with the URA3-loss assay (Supplementary notes). Therefore, we focused our follow-up analyses of whole-genome sequencing data on hybrids for which the RTG increased the LOH ratio.

**RTG recombination in hybrids with extensive chromosomal rearrangements**. We performed whole-genome sequencing of RTG-evolved clones derived from the fertile ScWE/ScNA hybrid ($n = 24$ T6) and the sterile ScMA/ScNA hybrid ($n = 123$ T6, $n = 2$ T4, $n = 3$ T0), which have similar levels of heterozygosity but different genome structures (Fig. 2b and Supplementary Data Set 9). The sterility in the ScMA/ScNA hybrid is driven by chromosomal rearrangements, with five homologous chromosomes for which the parental subgenomes are not collinear over long stretches. Our analyses revealed that the RTG genomes of ScWE/ScNA and ScMA/ScNA clones recombined similarly often (with 13.79 and 12.41 LOHs on average per clone, respectively; Fig. 3a–b and Supplementary Data Set 6). This underscores that genome-wide RTG recombination is not hampered by extensive non-collinearity between subgenomes, moreover, collinear and non-collinear chromosomes recombine with the same frequency in the ScMA/ScNA RTGs (Supplementary notes and Supplementary Data Set 15) and is consistent with sterility being caused by the missegregation of the non-collinear chromosomes, rather than by a lack of recombination between them[27]. In addition, we found that neither parental subgenome was favoured in terms of the homozygosity produced (Supplementary Fig. 2c).

In both hybrids, the genomic intervals encompassing the LOH breakpoints on the right arm of chromosome II had lower heterozygosity (Fig. 3c, Supplementary Fig. 4a–b). Moreover, the LOH breakpoints also coincided with known meiotic hotspots[28] and sites where Spo11p induces DSBs during meiosis[29] (Fig. 3d, Supplementary Fig. 4c, Supplementary Data Set 10). Since this implies that RTG recombination relies on DSBs produced during meiosis, we probed whether LOH breakpoints were associated with known meiotic recombination hotspots genome-wide and found this to be the case (Supplementary Fig. 4c, Supplementary

Data Set 11). This is in agreement with the occurrence of the Spo11p-induced DSBs at highly conserved sites across the S. cerevisiae and S. paradoxus lineages used here[30]. We also compared the relative intensities (RIs) of hotspots overlapping or non-overlapping LOH breakpoints (Methods) and found the former to be higher in all hybrids (Supplementary Data Set 11). We found that centromeres were always maintained in a heterozygous state, in accordance with the lower level of DSBs formation in pericentromeric regions and the mitotic-like segregation of RTG (Supplementary Fig. 1c). We found no increase in single-nucleotide variants (SNVs), indels, aneuploidies or copy number variants (CNVs) and concluded that RTG-induced recombination does not cause global genome instability (Supplementary Fig. 5a). This is in line with the error-free homologous recombination pathway playing a major role in repairing meiotic DSBs.

Four ScWE/ScNA and nine ScMA/ScNA RTG clones carried a much larger fraction of markers in LOH (15–34% and 14–35%, respectively) than the median (1% and 1.7%, respectively) (Supplementary Fig. 2c). We hypothesised that extensive LOH could make the two subgenomes sufficiently homozygous to alleviate the sterility of the ScMA/ScNA hybrid. Thus, we probed the capacity of three ScMA/ScNA RTG clones with extensive LOH to produce viable gametes. Indeed, these clones had up to threefold higher gamete viability compared to the ancestral ScMA/ScNA hybrid (Supplementary Fig. 6a and Supplementary Data Set 3), confirming that RTG can help restore meiotic fertility by homogenising subgenomes.

We also detected LOHs encompassing the MAT locus on chromosome III in two of the ScWE/ScNA and in one of the ScMA/ScNA extensively recombined clones (Supplementary Fig. 6b). MAT locus homozygosity is well known to prevent yeast from entering meiosis and indeed these clones were completely sterile. However, the MAT locus homozygosity also made these clones mating proficient (Supplementary Fig. 6c–d), in contrast to MAT locus heterozygotes that are mating-deficient. This shows how aborted meiosis followed by a return to asexual growth can provide a direct route to polyploidization (Supplementary Fig. 6c–d). Overall, these results demonstrate that sterile intraspecies hybrids can evolve through RTG bypassing the reproductive barriers caused by a divergent chromosomal structure, with even a single RTG cycle having a dramatic impact on genome evolution.

**Local homozygosity enables RTG recombination in interspecies hybrids**. To shed light on how extremely high heterozygosity shapes the RTG recombination landscape, we sequenced the genomes of evolved clones ($n = 57$ T6, $n = 5$ T12, $n = 4$ T24)

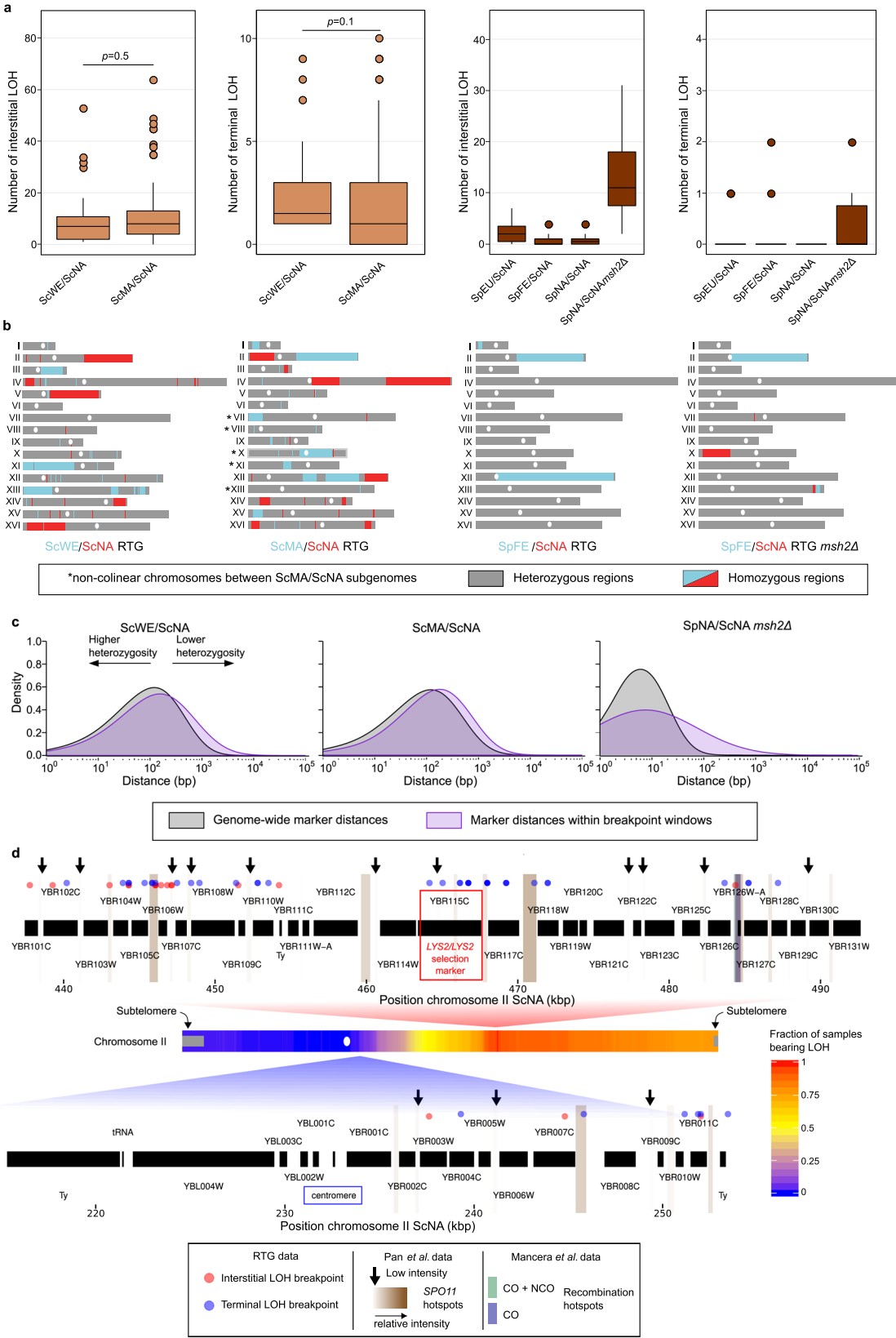

and non-evolved clones ($n = 28$ T0) derived from three inter-species *S. paradoxus*/*S. cerevisiae* hybrids (Supplementary Data Set 9). Their genome-wide RTG recombination was less extensive compared to intraspecies hybrids, generating fewer and smaller LOH (Fig. 3a–b, Supplementary Fig. 2, Supplementary Data Set S6). Nevertheless, 22 out of 34 wild-type RTG isolates had at

least one additional recombination event, besides the one selected on chromosome II. Moreover, one evolved clone, derived from the SpFE/ScNA hybrid, carried LOHs on three different chromosomes and spanning 9% of its genome (Fig. 3b). The latter suggested that highly recombined RTG clones do arise in inter-species hybrid populations, but at rather low frequencies. To test

**Fig. 3 LOH landscape of hybrids evolved through RTG. a** Left panels: boxplots of the number of interstitial and terminal LOHs in intraspecies hybrids. We detected no significant difference in the number of interstitial or terminal events comparing the ScWE/ScNA ($n = 24$) and the ScMA/ScNA ($n = 125$) data sets (two-sided Wilcoxon rank-sum test, $p$ value = 0.5 and $p$ value = 0.1, respectively). Right panels: boxplots of the number of interstitial and terminal LOHs in interspecies hybrids. Colour code as in Fig. 2a. **b** Genome-wide LOH patterns for two RTG-evolved intraspecies hybrids (left) and two evolved interspecies hybrids (right). All plots are based on the ScNA reference genome. **c** Distribution of the distances between consecutive markers, genome-wide (grey) and in LOH breakpoint windows (purple), across three different hybrids. LOH breakpoint windows comprise the five heterozygous markers and the five homozygous markers closer to the breakpoint. **d** Association between meiotic recombination and LOH breakpoints regions within two segments of chromosome II. LOH breakpoint regions are defined e.g. as the genomic interval between the first homozygous marker of a LOH region and the closest flanking marker which does not belong to the same LOH region. The colour intensity of Spo11p hotspots is proportional to the corresponding signal intensity. The heatmap of the LOHs (i.e., two copies of ScNA alleles) resulting from the breakpoints is also reported. The boxplot is defined as follows: the box is delimited by the first quartile (Q1) and the third quartile (Q3). The line that separates the box is the median. Whiskers are defined as: upper whisker = min(max($x$), Q3 + 1.5 × IQR); lower whisker = max(min($x$), Q1−1.5 × IQR), where: $x$ is the data, Q1 is the first quartile, Q3 is the third quartile and IQR is the interquartile range (IQR = Q3−Q1).

this possibility, we constructed a SpNA/ScNA hybrid carrying a second unlinked selectable heterozygous marker (*CAN1*), which enabled us to screen for an additional, independent LOH event. Genome sequencing of four RTG clones (T6) selected for loss of *URA3* and *CAN1* revealed clones with multiple LOHs, supporting the scenario that RTG can, although rarely, reshuffle highly heterozygous genomes (Supplementary Fig. 7).

We quantified the genome-wide effect of the mismatch-repair machinery in suppressing RTG recombination by sequencing SpNA/ScNA *msh2Δ* evolved and control clones, and found ~1000-fold increase in the median fraction of genome in LOH ($2.10 \times 10^{-4}$ vs $1.58 \times 10^{-7}$ in the wild-type) in samples isolated after the *URA3*-loss assay. Thus, although LOH formation was hampered by high levels of heterozygosity, the inactivation of the mismatch-repair system mitigated this effect. LOH breakpoints were characterised by low values of local heterozygosity compared with the genome-wide averages, underscoring that islands of local sequence homology facilitate RTG recombination also in highly heterozygous hybrids (Fig. 3c, Supplementary Fig. 4, Supplementary Data Set 7). The impact of a single RTG cycle on LOH formation was massive compared with the conventional vegetative growth, both in interspecies and intraspecies hybrids (Supplementary Fig. 5b). Besides the LOH formation, interspecies RTG clones were characterised by remarkably stable genomes (Supplementary Fig. 5a), with the only exception of a clone that carried an inversion underlying a complex CNV (Supplementary Fig. 5c). Overall, our results showed that RTG recombination in interspecies hybrids is strongly influenced by heterozygosity, and that inactivation of the mismatch-repair machinery mitigates the effect of high-sequence divergence.

**Mapping quantitative traits in a sterile hybrid.** We probed whether recombination induced by RTG can generate beneficial allelic combinations by measuring the fitness (population doubling time and yield) of 125 ScMA/ScNA recombined RTG clones and three T0 control samples across 82 environments (Supplementary Data Set 5). The fitness of a large subset of recombined RTG clones was often inferior compared to that of the non-evolved hybrid (Fig. 4a and Supplementary Fig. 8c). Moreover, in 33 out of the 45 environments for which phenotypic data for both parents were available, one or more clones also showed worse parent heterosis, i.e., it performed significantly worse than both parents (Supplementary Data Set 14). Nevertheless, all RTG recombinants were fitter than the non-evolved hybrids in at least some niches (Fig. 4a) and few showed best parent heterosis (Supplementary Data Set 14), i.e., their growth exceeded that of both parents. This supports that RTG can generate new beneficial allelic combinations that are capable of driving adaptation.

Since hybrid sterility precludes standard linkage or association analyses, we next explored whether recombination induced by

RTG provides a method for mapping causative genetic variants across lineages that are reproductively isolated post-zygotically. Recombinants that were homozygous for the Malaysian alleles at the *LYS2* locus suffered a severe fitness decline in four environments and became superior in two environments. Although we could not map the cause of this fitness variation more precisely, we connected fitness variability in 33 environments to 45 quantitative trait loci (QTLs) (Fig. 4b). We found the doubling time in 13 environments to be linked to CNVs resulting from the recombination between non-collinear segments of the North American chromosome VII and the Malaysian chromosome VIII (Fig. 4b–c, Supplementary Fig. 8). Recombinants missing the right arm of Malaysian chromosome VIII ($n = 7$) grew slower in all 13 environments, and this explained many (13/33) of the worse parent heterosis cases. In contrast, those missing the corresponding North American ($n = 2$) chromosome arm grew faster or equally fast as non-recombinant samples (Supplementary Fig. 8).

Next, we asked whether QTL mapping by RTG provides sufficient resolution to associate genetic variants to the traits of sterile hybrids. First, we focused on an arsenic resistance QTL located just before the subtelomere of the right arm of chromosome XVI (Fig. 4b and Supplementary Fig. 9), a region that harbours the *ARR* locus, a gene cluster controlling arsenic exclusion from the cell. The *ARR* cluster is absent in the ScMA[13] and all the recombinants that had lost the corresponding ScNA allele were highly sensitive to arsenic (Supplementary Fig. 9). Then, we explored a major QTL on chromosome XV associated to higher fitness in presence of the antifungal drug cycloheximide. Near the QTL peak, we found the transcription factor *YRR1* that mediates drug resistance. Since the Malaysian *YRR1* contains two amino-acid substitutions predicted to be deleterious (Fig. 4d, Supplementary Data Set 8), we tested whether *YRR1* drives variation in cycloheximide resistance by reciprocal hemizygosity. We found hemizygous strains carrying the Malaysian *YRR1* to be less fit than those carrying the North American allele (Fig. 4e). We conclude that although a single cycle of RTG does not provide the recombination levels of standard meiosis, the resolution of QTL mapping based on a single cycle of RTG can be sufficient to map the genetic factors underlying the traits of sterile hybrids.

## Discussion

We evolved 20 diploid yeast genetic backgrounds with varying levels of sterility through RTG, challenging the dogma that sterile hybrids are evolutionary dead ends. We showed that aborting meiosis and returning cells to mitotic growth allows sterile yeast hybrids to bypass the reproductive barriers due to mutational defects, structural differences between subgenomes as well as extreme levels of heterozygosity, and to produce viable and often

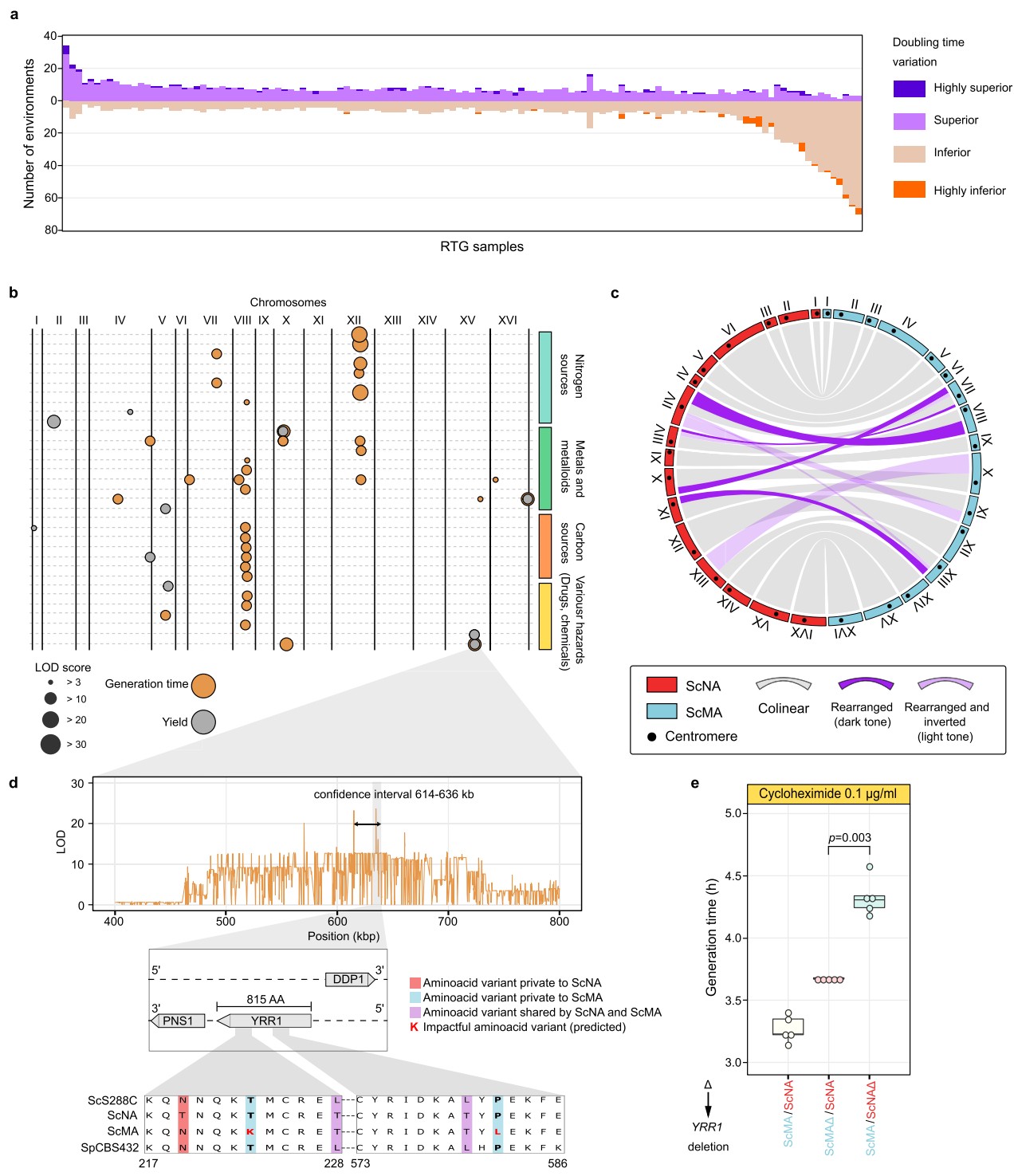

**Fig. 4 Fitness diversification of RTG clones and QTL mapping. a** Extent of fitness variation in the ScMA/ScNA data set. The *y* axis shows the number of environments in which the cell doubling time of the RTG clone is superior or inferior compared with the ScMA/ScNA parent hybrid. **b** QTLs mapped across the environments. Only conditions where QTLs were detected are reported. **c** Circular plot representing the rearrangements between the two subgenomes of the ScMA/ScNA hybrid. Regions are considered inverted if they do not resemble the ancestral centromere–telomere orientation. **d** Linkage scan for growth in media containing cycloheximide. Zoom-in on chromosome XV QTL and the two highly conserved regions of *YRR1* with two amino-acids substitutions predicted to be deleterious. **e** Boxplot of doubling times in cycloheximide for the parental ScMA/ScNA strains (five replicates) and the two hemizygous strains (five replicates). The deletion of either allele reduces growth in cycloheximide compared with the WT, underlying *YRR1* haploinsufficiency in this background. The deletion of the ScNA allele is significantly worse (one-tailed Wilcoxon rank-sum test *p* value = 0.003) than the ScMA. The boxplot is defined as follows: the box is delimited by the first quartile (Q1) and the third quartile (Q3). The line that separates the box is the median. Whiskers are defined as: upper whisker = min(max(*x*), Q3 + 1.5 × IQR); lower whisker = max(min(*x*), Q1–1.5 × IQR), where: *x* is the data, Q1 is the first quartile, Q3 is the third quartile and IQR is the interquartile range (IQR = Q3−Q1).

perfectly fit recombinant clones. These mechanisms constitute some of the post-zygotic reproductive barriers that often underlie speciation in nature, in yeast and other species[4].

We observed that the extent of RTG-induced recombination depends on two factors acting one after the other: the meiotic progression and the recombination efficiency, respectively. We investigated these factors with two metrics: the LOH difference and the LOH ratio. These metrics showed that the high heterozygosity of interspecies hybrids resulted in lower recombination levels compared with intraspecies hybrids. However, abolishing the mismatch repair by removing the mismatch-binding protein Msh2p promoted RTG recombination, mirroring the increased meiotic recombination observed in the gametes of similar interspecies hybrid[31].

The LOH regions generated by recombination between hybrid subgenomes can have profound evolutionary consequences. We had previously shown that the homozygous blocks produced by LOH can mediate meiotic recombination between highly diverged subgenomes, which in turn can rescue hybrid fertility and initiate interspecies introgressions in yeast[32]. Here, we showed that RTG-induced LOHs can make the mating type locus homozygous, thereby giving rise to mating-proficient diploid hybrids. This provides an obvious direct route to polyploidization that can restore the fertility of sterile hybrids by whole-genome duplication[33,34]. RTG-driven polyploidization would produce a similar outcome to what is observed in plants, where polyploidization can result from the mating of endoreplicated gametes with an unreduced genome content[35], and may help explain the abundance of yeast polyploids in nature[36]. LOH regions might be selected by adaptation under specific selective regimes[37–40] but may also be constrained by incompatibility between allele pairs located in different subgenomes[41]. In our experimental design, we minimised the selective pressure acting on the RTG samples as they were grown for a limited number of generations. This precluded LOH enrichment towards one of the hybrid subgenomes at specific loci, in contrast to what has been observed in adaptive evolution experiments[38,42]. Furthermore, the maintenance of a stable diploid state in RTG clones might promote unique evolutionary trajectories, such as the ability to tolerate large CNVs that would otherwise kill haploid gametes. RTG clones carrying such CNVs are however likely to be unfit in many environments, as underscored by a single large CNV that explained many of the cases of worse parent heterosis observed here. LOHs over regions containing recessive loss-of-function alleles also likely contribute to the worse parent heterosis[43]. This could counteract the masking of low-fitness alleles that appears to account for much of the pervasive mid- and best parent heterosis in highly heterozygotic yeast F1 hybrids[44–46]. By unmasking deleterious recessive alleles through LOH and exposing them to the selection, RTG may also serve as an evolutionary mechanism to purge negative variation from populations in a similar way as self-fertilisation is predicted to do[47]. In concert, the impact of RTG recombination on yeast natural evolution has the potential to be substantial. In fact, it is induced simply by brief starvation that is followed by cells encountering again nutrients, which may be an occurrence in fluctuating wild habitats[48]. Whether the role of RTG recombination also extends across broader swaths of the tree of life is unknown, but yeasts separated by hundreds of millions of years of evolution from *S. cerevisiae* experienced LOH with a pattern compatible with the RTG signature without a specific tailored protocol[49,50]. We see no evident mechanistic reasons why the RTG process would not extend also across more distantly related facultative sexual organisms in which meiosis is induced by starvation.

Mechanistic deviations from the meiotic paradigm have been repeatedly reported across the eukaryotic tree of life. A meiotic gene mutant that skips the second meiotic division and produces gametes with unreduced genomic content and complementary recombined genomes was reported in *Arabidopsis thaliana*[51]. Furthermore, other organisms such as *Candida albicans*[52,53] or the rotifer *Adineta vaga*[54] experience genetic recombination without a conventional sexual cycle. The recent characterisation of the lifecycle of more than 1000 *S. cerevisiae* strains revealed an independent loss of sexual reproduction in many lineages[8]. Population genomics data revealed pervasive genome-wide signatures of historical LOHs[55], suggesting that RTG might have contributed to the genome evolution of sterile yeast strains. The access to this para-meiosis provides a powerful alternative path for genome evolution that breaks the paradigm of yeast sterile hybrids as evolutionary dead ends.

## Methods

**Construction of hybrid yeast strains**. All the *Saccharomyces cerevisiae* (Sc) and *Saccharomyces paradoxus* (Sp) strains constructed and used in this study are reported in Supplementary Data Set 1. The diploid hybrid ScS288C/ScSK1 *ndt80Δ* was generated by mating the heterothallic (*ho*) ScS288C and ScSK1 commonly used laboratory strains both deleted for *NDT80* (*ndt80::KanMX*) and diploids complementing for histidine auxotrophy were selected. The haploid parental heterothallic (*ho::HygMX*) strains ScNA, ScWA, ScWE, ScSA, ScMA used for generating the intraspecies and interspecies hybrids were previously described[56,57]. In each Sc *MATa* and *MATα* haploid background, the native *URA3* on chromosome V was deleted (*ura3::KanMX*). Subsequently, the *URA3* gene was inserted at the *LYS2* locus in the *MATα* haploid background (*MATα, ura3::KanMX, lys2::URA3*). For generating the intraspecies hybrids, *MATa* cells and *MATα* cells of the two Sc parental species were mated and prototroph diploids (*MATa/MATα, ura3::KanMX/ura3::KanMX, LYS2/lys2::URA3*) were selected on minimal media lacking both lysine and uracil. The haploid parental heterothallic (*ho::HygMX*) strains SpEU, SpFE, SpNA were generated for this work following the genetic engineering scheme explained above for *MATa* and *MATα* strains. For generating the interspecies hybrids, *MATα* cells and *MATa* cells from the Sc and Sp parental species were mated and prototroph diploids (*MATa/MATα, ura3::KanMX/ura3::KanMX, LYS2/lys2::URA3*) were selected on minimal media lacking both lysine and uracil. The SpNA/ScNA *ndt80* and *msh2* diploid mutants were generated by deleting each gene in the respective Sc or Sp haploid background (*ndt80::NatMX* or *msh2::NatMX*) using the lithium acetate/PEG transformation protocol. The haploid strains were mated and diploid prototrophs (*MATa/MATα, ura3::KanMX/ura3::KanMX, LYS2/lys2::URA3, msh2::NatMX/msh2::NatMX*) or (*MATa/MATα, ura3::KanMX/ura3::KanMX, LYS2/lys2::URA3, ndt80::NatMX/ndt80::NatMX*) were selected on minimal media lacking both lysine and uracil. The *CAN1* deletion in the SpNA background was obtained using the lithium acetate/PEG transformation protocol and the diploid hybrid was obtained by mating with ScNA (*MATa/MATα, ura3::KanMX/ura3::KanMX, LYS2/lys2::URA3, can1:NatMX/CAN1*). All the deletions made were verified by polymerase chain reaction (PCR). All the PCR primers used in this study are reported in Supplementary Data Set 12.

***URA3*-loss LOH assay**. Each hybrid was patched from the −80 °C glycerol stock on YPD solid media (1% yeast extract, 2% peptone, 2% dextrose and 2% agar) and incubated overnight at 30 °C. The following day the strain was streaked to minimal solid media not supplemented with uracil and the plate was incubated at 30 °C for 48 h. Five single colonies of the hybrid strain were taken and inoculated separately in 10 mL of YPEG pre-sporulation medium (1% yeast extract, 2% peptone, 3% ethanol and 3% glycerol) for 15 h at 30 °C with shaking at 220 rpm. Each presporulation culture was washed twice with sterile water and resuspended in sporulation medium (2% potassium acetate) to reach an OD$_{600}$ = 0.5 in a 250-mL flask. One mL was immediately collected from the starving culture to generate the T0 sample (before meiosis induction). The flasks were incubated at 23 °C, with shaking at 220 rpm, and after 6 h of incubation the T6 samples were collected. For the ScNA/SpNA *ndt80* mutant, two additional time points were collected after 12 and 24 h of incubation (T12 and T24, respectively). The T0, T6, T12 and T24 samples were washed twice with 1 mL of YPD and incubated in 1 mL of nutrient-rich YPD for 18 h at 30 °C without shaking, thereby aborting meiosis in the samples and RTG. The following day the YPD liquid cultures were vortexed and, depending on the strain and the density of the culture, 20 to 200 μL of 10-fold diluted culture were plated on a minimal medium containing 1 mg/mL of 5-fluoroorotic acid (5-FOA) and spread with glass beads[39]. In parallel, cells from the same YPD liquid culture were serially diluted up to 10$^{-5}$ and spotted on YPD plates in at least two replicates for each biological replicate. The YPD and 5-FOA plates were then incubated at 30 °C for 48 h. After that, colonies growing on 5-FOA (and therefore having lost their *URA3* marker through LOH) and YPD plates were counted and used to calculate the colony-forming units per mL for T0, T6, T12 and T24 samples plated in both 5-FOA and YPD. Finally, we calculated the LOH rate at

the different time points according to the equations:

$$R_{T0} = 100 \cdot (CFU_{5-FOA,0}/CFU_{YPD,0}),$$

$$R_{T6} = 100 \cdot (CFU_{5-FOA,6}/CFU_{YPD,6}),$$

$$R_{T12} = 100 \cdot (CFU_{5-FOA,12}/CFU_{YPD,12}),$$

$$R_{T24} = 100 \cdot (CFU_{5-FOA,24}/CFU_{YPD,24}),$$

where e.g.:

CFU$_{5-FOA,6}$ is the number of colony-forming units per mL of sample plated on 5-FOA after 6 h of sporulation induction followed by the induction of RTG in YPD as described above,

CFU$_{YPD,6}$ is the number of colony-forming units per mL of sample plated on YPD after 6 h of sporulation induction followed by the induction of RTG in YPD as described above.

The RTG rates were used to calculate the LOH ratio ($R_{T6}/R_{T0}$) and the LOH difference ($R_{T6}-R_{T0}$).

RTG clones from the S288C/SK1 hybrids were obtained using the mother–daughter protocol as previously described[17].

**Meiosis dynamics and spore viability**. The meiosis of diploid hybrids engineered with the *LYS2/lys2::URA3* system was induced with the same protocol used for RTG but the flasks were kept at 23 °C with shaking at 220 rpm and monitored for the formation of spores. In order to analyse the sporulation efficiency (fraction of cells having passed through meiosis), at least 200 cells were counted for each sample and estimated the percentage of those that had sporulated (formed dyads, triads or tetrads of spores). To estimate spore viability (gametes that are viable and capable of germinating), complete tetrads were dissected and their spores were deposited on YPD plates where they were allowed to germinate and grow for four days at 30 °C. The spore viability was calculated as the fraction of spores that had germinated and formed visible colonies. To monitor meiotic progression by DAPI staining 2 mL of T0, T6 and T12 samples were put in a separate tube with 5 mL of EtOH 70% and stored at −20 °C. Frozen samples were washed twice with 1 mL of sterile water, resuspended in 200 μL of water and stained with 2 μL of DAPI (4′,6-diamidino-2-phenylindole) for 30 min in the dark. Cells were counted using a fluorescence microscope.

**RTG selection by *URA3*-loss assay and *URA3/CAN1*-loss assay**. We designed the 5-FOA assay to detect the increase of recombination upon RTG at the heteroallelic locus *LYS2/URA3* on chromosome II. We deleted the copy of *URA3* from its native location on chromosome V in all the haploid parental strains. We replaced one *LYS2* allele on chromosome II with one copy of *URA3* in one of the two parents used to generate each diploid hybrid, and performed the assay as already reported in the literature[24]. The growth in YPD for 18 h ensured that all the cells that were in the early phase of meiosis performed RTG. We used the same incubation time in liquid YPD for the sporulating cultures (T6) and the controls (T0) to take into account LOH occurring during the growth in YPD. We confirmed by whole-genome sequencing of single clones that de novo mutations and aneuploidies did not impact the heteroallelic assay, and all events that lead to *URA3* loss were LOHs. Therefore, we concluded that T0 cells growing on 5-FOA were due to mitotic LOH, whereas T6 cells had a composite effect of mitotic and RTG-induced recombination. Sequencing of single clones supported this scenario, and T6 RTGs had a single LOH event on chromosome II (mitotic recombination or low efficiency RTG), or additional LOHs (RTG) in the genome. This difference must be carefully considered also for interspecies hybrids, although in those backgrounds the lack of additional LOH events can result from anti-recombination mechanism due to the high-sequence divergence.

We expanded this protocol to introduce an additional selection step as already reported in the work by Coelho et al. in which multiple selection steps were used[58] and also based on an early work in which RTG cells had recombination at unlinked genetic markers[59]. We introduced an additional selection marker by deleting one copy of *CAN1* on chromosome V in one of the two parents of the hybrid tested, so that two independent LOHs could be selected for, one on chromosome II, at the *LYS2/ lys2::URA3* locus, and one on chromosome V, at the *CAN1/can1::NatMX* locus. In brief, after 18 h of YPD incubation, RTG cells were diluted 1:10 to an OD of ~0.5 in SD-based medium (0.675 % YNB, 0.0875% Arginine drop out, 2% Dextrose) supplemented with canavanine (2%), and grown for 10 h. Finally, cells were plated on 5-FOA plates following the protocol described in the paragraph "*URA3*-loss LOH assay" in the Materials and Methods section. The canavanine liquid incubation was enriched for cells bearing LOH at the *CAN1* locus and the passage on 5-FOA plates selected for cells carrying LOH at the *LYS2* locus.

**Sequencing data analysis**. All the samples were sequenced with Illumina paired-end technology at the NGS platform of Institut Curie according to the manufacturer's standard protocols. Short-read sequencing data were processed by means of the MuLoYDH pipeline using default parameters and the assemblies/

annotations embedded in MuLoYDH[24]. All the data sets consisted of the hybrids evolved under the RTG protocol and analysed against the corresponding parent hybrid before RTG as control samples (Supplementary Data Set 9). The pipeline required as input: (1) a data set of short-read sequencing experiments from yeast diploid hybrids and (2) the two parental genomes which were used to produce the hybrids in FASTA format as well as the corresponding genome feature annotations in the "general feature format" (GFF). The availability of reference-quality genome assemblies for all the parents used to generate the panel of hybrids, allowed a highly accurate tracking of the mutational landscape. Reads from sequencing data were mapped against the assemblies of the two parental genomes separately (standard mappings) and against the union of the two aforementioned assemblies (namely a multi-FASTA obtained by concatenating the two original assemblies) to produce the competitive mappings. In the latter case, reads from parent 1 were expected to map to the assembly of parent 1 on the basis of the presence of single-nucleotide markers. Conversely, reads from parent 2 were expected to map to the assembly of parent 2.

Standard mappings were used to determine the presence of CNVs. The latter were also exploited to discriminate LOHs owing to recombination from those resulting by deletion of one parental allele. The markers between the parental assemblies were determined by the NUCmer algorithm (version 4.0.0 beta) and were exploited to map LOH segments. Markers were genotyped from standard mappings. Marker positions characterised by non-matching genotype or alternate allele were filtered out, as well as multiallelic sites, whereas those lying in subtelomeric and telomeric regions were masked. Remarkably, MuLoYDH provided LOH calls without filtering out small events using an arbitrary threshold based on the number of supporting markers. Indeed, we were able to detect events supported by a single marker and we previously demonstrated that such events are genuine LOH[24]. Stretches of consecutive markers showing homozygous genotypes were grouped in LOH regions. The genomic coordinates of each LOH event were determined using both the "first/last" coordinates and the "start/end" coordinates. First/last coordinates were determined using the coordinates of the first and the last markers of the event. Start/end coordinates were calculated using the average coordinate of the first (last) marker and the last (first) marker of the adjacent event. LOH regions were annotated as terminal/interstitial. Interstitial LOHs were defined as homozygous segments flanked on both sides by heterozygous markers. Terminal LOHs were defined as homozygous regions extended to the end of the chromosomal arm, i.e., the last non-masked marker.

We excluded two WT RTG derived from ScS288C/ScSK1 owing to low coverage that did not match the standard requested by MuLoYDH. We also removed one sample from the WT ScS288C/ScSK1 RTG data set that went through two cycles of RTG in order to have only samples that performed one RTG cycle in WT and *ndt80Δ*.

De novo SNVs and indels were determined from both competitive and standard mappings. Competitive mapping allowed for direct variant phasing in heterozygous regions. Variant calling from competitive mapping was performed setting ploidy = 1 in heterozygous regions and ploidy = 2 in LOH blocks. Regions characterised by reads with low mapping quality (MAPQ < 5 in the competitive mapping) were assessed from standard mapping using arbitrarily the assembly from parent 1. All the de novo variants detected were checked by visual inspection using IGV[60].

**Testing for association between LOH breakpoints and Spo11p-induced DNA DSB or recombination hotspots**. The association between LOH breakpoints (using the start/end coordinates) and DSB/recombination hotspots was tested using the regioneR package[61]. In brief, regioneR implements a permutation test framework specifically designed for testing the association between two sets of genomic regions. The association tests were performed by means of the function *overlapPermTest*, setting the parameters: ntimes = 10000, and alternative = "greater". The fasta suite[62] (fasta36 -b 3 -d 3 -m 8, version: 36.3.8d April, 2016) was used to convert the coordinates of the hotspots regions detected in the original reference genomes (SGD_R62-1-1_20090218 and SGD_R58-1-1_20080305 for the hotspots reported by Pan et al.[29] and Mancera et al.[28], respectively) to the corresponding coordinates of the hybrid genome.

LOH breakpoint regions are defined, e.g., as the genomic interval between the first homozygous marker of an LOH region and the closest flanking marker which does not belong to the same LOH region.

Moreover, regioneR also implements a function to evaluate the local specificity of the detected association. It allows calculating the Z score of the statistical test for LOH breakpoints and recombination hotspots association as a function of the shift (in bp) of the recombination hotspot regions. In brief, the statistical test is performed iteratively changing the position of the recombination hotspot regions within a window of width $W = 10 \cdot S_m$, where $S_m$ is the mean size of the hotspot regions (transformed in the genomic coordinates of the hybrid as described above). For each iteration, the coordinates are shifted adding a step $T = S_m/2$. The procedure is then repeated subtracting step T to the coordinates of the hotspots.

We also compared the intensity (provided by Pan et al.[29]) of the hotspot regions (Spo11p-induced DNA DSBs) that overlap the LOH breakpoints (overlapping hotspots) against the intensity of the hotspot regions that do not overlap LOH

breakpoints (non-overlapping hotspots). The relative intensity (RI) was calculated as the ratio between the number of hits (H, as reported in the original study) and the maximum hit value: $RI = H/max(H)$. The test was performed using different padding values (0, 500, 1000, 2000 and 5000 bp) to define the LOH breakpoint boundaries, which were compared with the hotspots (Supplementary Data Set 11).

**Genome content analysis of polyploid strains**. The DNA content of the mating-proficient diploid RTGs was analysed upon mating with tester strains using propidium iodide (PI) staining assay. Cells were first pulled out from glycerol stocks on YPD solid media and incubated overnight at 30 °C. The following day a small portion of each patch was taken with a pipette tip, transferred in 1 mL of liquid YPD and incubated overnight at 30 °C. Then, cells were washed with water, resuspended in 1 mL of cold 70% ethanol and fixed overnight at 4 °C. Finally, the samples were washed twice with phosphate-buffered saline (PBS), and 100 μL of each sample were resuspended in 900 μL of staining solution (15 μM PI, 100 μg/mL RNase A, 0.1% v/v Triton-X, in PBS) and incubated for 3 h at 37 °C in the dark. Ten thousand cells for each sample were analysed on a FACS-Calibur flow cytometer. Cells were excited at 488 nm and fluorescence was collected with a FL2-A filter. The data were analysed using the R packages flowCore[63] and flowViz[64], and were plotted with ggplot2 (R version 3.6.1).

**Long reads sequencing and structural variant analysis**. Yeast cells were grown overnight in liquid YPD media. Genomic DNA was extracted using Qiagen Genomic-Tips 100/G according to the manufacturer's instructions. The MinION sequencing library was prepared using the SQK-LSK109 sequencing kit, according to the manufacturer's protocol. The library was loaded onto a FLO-MIN106 flow cell and sequencing was run for 72 h. Long-read basecalling and scaffolding were performed using the LRSDAY pipeline (version 1.6)[65] and the dot plot of chromosome III was generated using mummerplot[66]. The inversion on chromosome III was detected using sniffles[67] implemented within the Varathon framework (https://github.com/yjx1217/Varathon).

**Estimating growth during mitotic reproduction**. All yeast strains were stored at −80 °C in 20% glycerol and cultivated at 30 °C in temperature and humidity-controlled cabinets. Yeast strains were revived from frozen 96-well stocks by robotic transfer (Singer RoToR; long pins) of a random sub-sample of each thawed population (~50,000 cells) to a Singer PlusPlate in 1536 array format on solid Synthetic Complete (SDC) medium composed of 0.14% Yeast Nitrogen Base (CYN2210, ForMedium), 0.50% (NH₄)₂SO₄, 0.077% Complete Supplement Mixture (CSM; DCS0019, ForMedium), 2.0% (w/w) glucose, pH buffered to 5.80 with 1.0% (w/v) succinic acid and 0.6% (w/v) NaOH. In every fourth position, fixed spatial controls (genotype: YPS128, MATa/MATα) were introduced to account for spatial variation across plates in the subsequent experimental stage. Controls were similarly subsampled from a separate 96-well plate and introduced to the pre-culture array (Singer RoToR; long pins). Populations were pre-cultivated for 72 h at 30 °C. For pre-cultures of nitrogen-limited environments, the background medium was modified to avoid nitrogen storing and later growth on stored nitrogen: CSM was replaced by 20 mg/L uracil (not converted into usable nitrogen metabolites) and (NH₄)₂SO₄ was reduced to growth-limiting concentrations (30 mg/L of nitrogen). We cast all solid plates 24 h prior to use, on a levelled surface, by adding 50 mL of medium in the same upper right corner of the plate. We removed excess liquid by drying plates in a laminar air-flow in a sterile environment. Pre-cultured populations were mitotically expanded until stationary phase (2 million cells; 72 h), were again subsampled (~50,000 cells; short pins) and transferred to experimental plates, containing the medium of interest (Supplementary Data Set 5). Synthetic grape must was prepared as previously described. We tracked population size expansion using the Scan-o-Matic system, version 1.5.7 (https://github.com/Scan-o-Matic/scanomatic)[68]. Plates were maintained undisturbed and without lids for the duration of the experiment (72 h) in high-quality desktop scanners (Epson Perfection V800 PHOTO scanners, Epson Corporation, UK) standing inside dark, humid and thermostatic cabinets with intense air circulation. Images were analysed and phenotypes were extracted and normalised against the fourth position controls using Scan-o-Matic. We extracted the normalised, relative population size doubling time, $D_r$, by subtracting the control value for that position, $D_r = \log_2(D) - \log_2(D_{control,local})$. $D_r$ is reported as output data. Growth phenotypes were qualitatively classified as inferior, when $D_r$ normalised to the initial hybrid was between 0.25 and 1, highly inferior when $D_r$ was ≥1, superior when $D_r$ was between −0.25 and −1 and highly superior when the $D_r$ was ≤−1.

**QTLs mapping and phenotype analysis**. QTL analysis was performed with the package R/qtl (version 1.46-2)[69] using mitotic growth data normalised with the growth data of the parental hybrid for 125 RTG clones and 3 T0 samples of the ScNA/ScMA hybrid as the phenotype variable and the genotype of heterozygous markers extracted from the sequence data of the same RTG clones as the genotype variable. We removed markers from the data set to which the pipeline did not assign any genotype with the function "drop.nullmarkers", and we also removed markers for which only one sample was found to be homozygous. We estimated a genome-wide LOD statistical threshold performing 1000 random permutations of both the phenotype and the genotype rows and extracting the 95th percentile of

LOD values as threshold. The confidence interval of the QTL was calculated using the "lodint" function of R/qtl[69]. Best parent heterosis and worst parent heterosis were calculated as in Zörgö et al.[44]

**Reciprocal hemizygosity assay and functional mutation prediction**. The start-to-stop deletion of YRR1 was engineered in the parental haploid strains CC407 (ScNA) and YGL1027 (ScMA) by replacing the open reading frame with the NatMX cassette. The haploid strain CC407 yrr1::NatMX was then crossed with a wild-type haploid YGL1027 to obtain a hemizygous diploid carrying only the ScMA YRR1 allele. The haploid YGL1027 yrr1::NatMX strain was crossed with a wild-type haploid CC407 to obtain a hemizygous diploid carrying only the ScNA YRR1 allele.

Cells from the wild-type ScMA/ScNA hybrid and the two hybrids reciprocally hemizygous for YRR1 were pre-grown in a 96-well-plate containing YPD for 16 h in five biological replicates for each strain background. The following day, 20 μL of cultures were taken from each well and transferred to another 96-well-plate containing 180 μL of YPD with cycloheximide (0.1 μg/mL). The growth was monitored by measuring OD changes over 72 h on a Tecan plate reader (infinite F-200 pro). The generation time was extracted from the OD profile using the software PRECOG[70] and then plotted and analysed in Rstudio. The prediction of deleterious variants was obtained from the mutfunc suite[71].

**Reporting summary**. Further information on research design is available in the Nature Research Reporting Summary linked to this article.

## Data availability

The genome sequences generated in this study are available at Sequence Read Archive (SRA), NCBI under the accession codes: PRJNA681162. The phenotype data are available within the supplementary information files. Source data are provided with this paper.

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

## Acknowledgements

This work was supported by Agence Nationale de la Recherche (ANR-11-LABX-0028-01, ANR-13-BSV6-0006-01, ANR-15-IDEX-01, ANR-16-CE12-0019 and ANR-18-CE12-0004), Fondation pour la Recherche Médicale (FRM EQU202003010413), CEFIPRA, Cancéropôle PACA (AAP Equipment 2018), Meiogenix and the Swedish Research Council (2014-6547, 2014-4605 and 2018-03638). S.Mo. is funded by the convention CIFRE 2016/0582 between Meiogenix and ANRT. The Institut Curie NGS platform is supported by ANR-10-EQPX-03 (Equipex), ANR-10-INBS-09-08 (France Génomique Consortium), ITMO-CANCER and SiRIC INCA-DGOS (4654 programme).

## Author contributions

A.N. and G.L. conceived the project; S.Mo., L.T., J.W., A.N., G.L., designed the experiments; S.Mo., L.T., A.Ll., A.I., B.P.B., N.Š., M.D.C., J-X. Y., M.J. D., A.Lu., S.L, R.L., S.Ma., S.S., K.P., performed and analysed the experiments; B.A., performed the sequencing, J.L.L., S.D., J.W., A.N., G.L., contributed with resources and reagents; S.Mo., J.W., A.N., G.L. supervised the project; G.L. coordinated the project; S.Mo. and G.L. wrote the paper with input from L.T., A.N., J.W.

## Competing interests

A.N. and G.L. have a patent application on "Yeast strains improvement method" using RTG (US20150307868A1). All the other authors declare no competing interests.
