## [Peer Review File · Nature Communications]

Title: Aborting meiosis allows recombination in sterile diploid yeast hybridsREVIEWER COMMENTS

Reviewer #1 (Remarks to the Author):

Hybrid diploids derived from parents that are diverged from each other are often sterile, and developing a means to overcome this barrier would be very useful in genetic analysis of traits and developing new allele combinations in a wide variety of organisms. In this paper, Mizzachiodi and coworkers report that allowing cells to enter meiosis and then returning them to the mitotic cell cycle (return to growth, RTG) allows the recovery of viable diploids that have recombined their genomes. Using both whole-genome sequencing and a simple loss-of-heterozygosity (LOH) assay based on loss of a URA3 gene, they examine the impacts of sequence divergence, genome rearrangement, and other factors on the recovery of recombinants using this protocol, and also show that the diploids emerging from RTG can be used to increase hybrid fertility, to generate genetic diversity, and to map traits under circumstances where very few viable haploid gametes are recovered. The approach and potential conclusions are of considerable interest, but there is a serious flaw in one of the methods used to compute LOH rates that needs to be corrected, and that has the potential to alter some of the major conclusions of the paper.

Major Comment: There is a need for significant reanalysis of the data derived from the simple genetic assay. The metric used throughout, called “LOH rate”, is a ratio of the frequency of Ura⁻ cells at t=0 and t=6. However, it is not clear what biological significance it has. One presumes that the f(Ura⁻) at t0 reflects a combination of the rate of mitotic recombination and of mitotic chromosome loss, while the f(Ura⁻) at t6 reflects a rate of meiotic recombination. This is literally comparing apples and oranges. It would be much more accurate to consider f(Ura⁻) at t0 to be background signal, as it reflects the background mitotic level as cells enter into meiosis, and to subtract it from f(Ura⁻) at t6. Please consider this seriously.

Even if one does this, it is not clear that the number derived has much meaning with regards to the recombination process, as the value will be dominated by the fraction of cells entering meiosis by 6h and by rates of meiotic progression (i.e. fraction of cells that initiate recombination by 6h). Diploids that enter meiosis poorly are going to give low meiotic LOH values, as authors point out. Not presenting this information in Figure 2D could lead to the wrong impression, particularly because when the two f(Ura⁻) are very small, they can give a large ratio but of course the difference will be equally small (see graph in attachment, LOH_comparison.pdf)

I would recommend a) reporting background subtracted meiotic LOH frequencies (i.e. differences) and b) reporting the frequencies of MI+MII at 12h below each column, so that the reader can figure out which values are low because of poor sporulation. Similarly, the plot in Figure 2E should report background-subtracted meiotic LOH frequencies, not ratios.

This incorrect reporting of LOH can lead to misleading conclusions. For example, ScNA/ScNA and ScMA/ScNA have very similar LOH “ratios”, but the background-subtracted meiotic LOH frequencies are almost 10-fold different: 1.9% versus 0.27%. In another example, SpNA/ScNA MSH2 and SpNA/ScNA msh2Δ show a 2-fold difference in background-subtracted LOH ratios, but show similar LOH ratios (8 versus 5) Therefore, the conclusion that non-colinearity does not reduce LOH (line 131) is incorrect, and

the impact of mismatch repair is modest, at best, using this assay (however, it appears to have a substantial impact when assayed by whole-genome sequencing). Many other conclusions may change, as well.

Line 71—it remains unclear whether or not the Spo11 complex actually has topoisomerase activity. Perhaps “a homolog of DNA topoisomerase VI”.

Line 76—“bud like mitotic cells”

Line 94—perhaps it is worth explaining that, the wild-type 4-5 h sample contains cells at different stages of meiosis, with different levels of meiotic DSBs, which would explain the heterogeneity of this population with regards to crossover numbers. This is stated later (line 104) but in a non-obvious way.

Line 137, Table S4. It was difficult to figure out which SpNA/ScNA hybrid was wild-type, *msh2Δ*, or *ndt80Δ* in Table S4. It would be useful to add this information.

Line 158, Table S9. It would be useful to include extent of sequence divergence and number of departures from collinearity for the hybrid strains listed in Table S9.

Line 170 and ff, Fig 3D and S4C. The association between breakpoints and DSB hotspots is presented in a way that is confusing and does not make the case clearly, and the section in the materials and methods is not very helpful. I am not sure how to improve this, but perhaps one could take the distribution of Spo11 oligo intensities in a region around each LOH breakpoint (suggest 2 kb on either side, but other distances might be tried) and compare it to the distribution of Spo11 oligo intensities genome-wide? Figure 2—The colors used in this figure are not so easy to differentiate from each other. Rather than numbering each class 1, 2, 3, 4 etc., authors might want to give each hybrid a unique number (i.e. homozygous parents 1,2,3,4; intraspecies hybrids 5,6,7, etc.

Table S13—please provide a legend to explain what the headings in “markers distribution per class” mean. In particular, what is meant by homozygous reciprocal, homozygous non-reciprocal, and homozygous 4:0? In general, the clarity of the supplementary tables would be considerably enhanced by the inclusion of explanatory legends that define each of the column headers.

Reviewer #2 (Remarks to the Author):

This is an impressive and methodologically thorough study providing interesting and novel content. I applaud the authors on this in-depth investigation of the return-to-growth (RTG) process in *Saccharomyces* yeasts, combining a large range of state-of-the-art approaches in yeast genetics, including laboratory hybrid crosses, genetic engineering, WGS, genomics analyses, high throughput phenomics, and QTL analysis. The authors show that normally sterile hybrid yeast strains can abort meiosis and return to mitotic growth, with the effect that a level of genomic recombination is maintained but reproductive species barriers are bypassed and fertility is (partly) restored in hybrids. The authors claim that this may represent a viable route to increased fitness and adaptation of yeast hybrids and other microbial eukaryotes that can reproduce both asexually and sexually.

While this study is certainly very interesting for a specialist audience, I think the authors may overstate the relevance of their findings a little, considering the broad range of readers of this journal. The return-to-growth mechanism is not likely to have the same wide-ranging effects on adaptive evolution in obligate sexual organisms with a more complex organisation and development. It would thus be preferable if the implications, especially with respect to hybrid speciation in the abstract and discussion, would be given some more qualifying conditions.

The other point I think the authors are overstating is the recombination rate increase, or the size of the RTG effect on LOH. It seems to me that this effect is highly cross dependent (which should be tested) and may only confer a slight increase compared to mitosis (Figure 2D). I don't think the effect is directly comparable to outcomes of sexual recombination, which includes segregation and causes new combinations of alleles within genomes.

Here are some more specific comments:

1. Have you tested whether recombination rates were correlated to sequence-wide genetic distance between parental strains in your data?
2. Line 124: The text mentions these cross abbreviations here without explanation. It took me a while, and a look at Figure 2, to realize that the ScMa/ScNA hybrid cross involves the highly non-collinear Malaysian strain. It would be good to mention this interesting detail more explicitly in the main text.
3. Line 137-142: I see no details of the *msh2*-deleted strain in Figure 2D or 2E at all, although the text refers to it.
4. Figure 1D. Is there a better way to display the mean/median and variance between clones to compare between these two plots? The Figure is not very convincing with respect to the point you are trying to make in the text (line 103 “the distribution was more homogeneous in the *ndt80Δ* population compared to the wild-type population.”)
5. Line 162: “This underscores that genome-wide RTG recombination is not hampered by extensive non-collinearity between subgenomes, in line with that the sterility caused missegregation of the non-collinear chromosomes rather than a lack of recombination.” I don't understand the second half of this statement. In my understanding, antirecombination causes missegregation, which in turn causes sterility (i.e. highly aneuploid, unviable gametes) of F1 hybrids. So in my mind, the causality is the other way around than you imply here.
6. Line 165: “Moreover, neither parental subgenome was favoured over the other in terms of the created homozygosity”. I am not suggesting you adding more work here but I am really intrigued why the two subgenomes are so well balanced in your hybrids. Why are they not more affected by the selective environment and/or epistasis as seen by others before (Smukowski Heil et al, MBE, 2017; Zhang & Bendixsen et al MBE 2019)? You are alluding to it briefly in line 276 (“LOH regions might be selected by adaptation under specific selective regimes but may also be constrained by

incompatibility.”) Would you expect an interaction between homozygosity and the environment here, e.g. do you expect to find larger homozygosity of the *S. cerevisiae* subgenome in media where the *S. cerevisiae* parent has higher fitness than *S. paradoxus*?

7. It would be good to see these results discussed in a more quantitative genetics context, e.g. considering the fact that many high fitness alleles have been found to be recessive and are only expressed in the homozygous state (Zörgö et al. MBE 2012; Bernardes et al. JEB, 2017). Do the authors consider their results to be important with respect to the role of dominance, overdominance or underdominance in hybrid fitness (e.g. as seen in Laiba et al. Genome, 2016)?

8. Legend to Figure 2, line 456: “...panels B, E and E”.

9. Update Bozdogan et al reference (now published in Current Biology <https://doi.org/10.1016/j.cub.2020.12.038>)

Reviewer #3 (Remarks to the Author):

This is an interesting and thorough paper showing that inducing meiosis by starving yeast cells, but then restoring nutrients before meiosis progresses, causes homologous recombination between the diploid chromosomes resulting in loss of heterozygosity (LOH, i.e. gene conversion). Normal mitotic recombination also produces such LOH events during normal diploid growth, but the return to growth method (RTG) greatly increases the rate, as shown in Figure 2D. This is a clever idea, and a potentially useful method for mapping phenotypes in sterile heterozygotes, and the authors prove the principle with a couple of nice experiments. Overall, the work is interesting and useful and clearly presented.

However, I have one major criticism which I think prevents it from being published in Nature Communications. The authors present the RTG technique as though it overcomes hybrid sterility, for example in the title, but it doesn't. What is usually meant by hybrid sterility is the inability of a hybrid to complete its sexual cycle successfully. Sex comprises the production of gametes (meiosis) followed by the fusion of gametes (syngamy). Sex has two genetic effects, segregation and recombination, which together can increase or decrease the genetic diversity within the diploid genomes of individuals of the next generation, and within the populations they comprise. Hybrid sterility in yeast usually means the inability of hybrid diploids to produce viable haploid gametes by meiosis, and this is what the authors claim to have overcome.

The authors consider three forms of sterility, e.g. in the introduction and in Figure 1. The first is the loss of function of meiotic genes, so that meiosis doesn't progress (this isn't strictly “hybrid sterility” since it is likely to affect non-hybrids too, unless the authors are suggesting that hybrids are more susceptible because of incompatibility between meiosis gene allele from different species which is plausible, but not made explicit). The second is non-colinearity between different species' chromosomes, which have

rearrangements relative to each other, so that meiosis produces inviable gametes lacking essential parts of chromosomes. And the third is anti-recombination, the inability of diverged chromosome to recombine, which causes meiotic chromosome mis-segregation and inviable gametes that lack essential chromosomes. The authors show that in each of these types of sterile hybrids, RTG increases recombination between chromosomes, so that diploids with LOH are produced. However, this is not the equivalent of successfully overcoming hybrid sterility. Sex produces gametes which can outcross, increasing genetic diversity and producing novel combinations of alleles. RTG doesn't produce gametes, and LOH only results in the loss of genetic diversity, without producing novel combinations. RTG does some of the things that sex does, but it's not, in my opinion, "overcoming hybrid sterility".

Response letter

We would like to thank reviewers for their constructive feedback and we have changed the manuscript to accommodate the reviewers' viewpoints. As suggested, we now report both LOH differences and LOH ratio metrics and use both to interpret the results throughout the text. A point-by-point response to the reviewers' comments ("replies" [R]) and a detailed description of the corresponding changes made to the manuscript ("actions" [A]) are reported below.

Reviewer #1 (Remarks to the Author):

Hybrid diploids derived from parents that are diverged from each other are often sterile, and developing a means to overcome this barrier would be very useful in genetic analysis of traits and developing new allele combinations in a wide variety of organisms. In this paper, Mozzachiodi and coworkers report that allowing cells to enter meiosis and then returning them to the mitotic cell cycle (return to growth, RTG) allows the recovery of viable diploids that have recombined their genomes. Using both whole-genome sequencing and a simple loss-of-heterozygosity (LOH) assay based on loss of a URA3 gene, they examine the impacts of sequence divergence, genome rearrangement, and other factors on the recovery of recombinants using this protocol, and also show that the diploids emerging from RTG can be used to increase hybrid fertility, to generate genetic diversity, and to map traits under circumstances where very few viable haploid gametes are recovered. The approach and potential conclusions are of considerable interest, but there is a serious flaw in one of the methods used to compute LOH rates that needs to be corrected, and that has the potential to alter some of the major conclusions of the paper.

Major Comment: There is a need for significant reanalysis of the data derived from the simple genetic assay. The metric used throughout, called "LOH rate", is a ratio of the frequency of Ura- cells at $t=0$ and $t=6$. However, it is not clear what biological significance it has. One presumes that the $f(\text{Ura-})$ at t_0 reflects a combination of the rate of mitotic recombination and of mitotic chromosome loss, while the $f(\text{Ura-})$ at t_6 reflects a rate of meiotic recombination. This is literally comparing apples and oranges. It would be much more accurate to consider $f(\text{Ura-})$ at t_0 to be background signal, as it reflects the background mitotic level as cells enter into meiosis, and to subtract it from $f(\text{Ura-})$ at t_6 . Please consider this seriously.

Even if one does this, it is not clear that the number derived has much meaning with regards to the recombination process, as the value will be dominated by the fraction of cells entering meiosis by 6h and by rates of meiotic progression (i.e. fraction of cells that initiate recombination by 6h). Diploids that enter meiosis poorly are going to give low meiotic LOH values, as authors point out. Not presenting this information in Figure 2D could lead to the wrong impression, particularly because when the two $f(\text{Ura-})$ are very small, they can give a large ratio but of course the difference will be equally small (see graph in attachment, LOH_comparison.pdf)

I would recommend a) reporting background subtracted meiotic LOH frequencies (i.e. differences) and b) reporting the frequencies of MI+MII at 12 h below each column, so that the reader can figure out which values are low because of poor sporulation. Similarly, the plot in Figure 2E should report background-subtracted meiotic LOH frequencies, not ratios.

This incorrect reporting of LOH can lead to misleading conclusions. For example, ScNA/ScNA and ScMA/ScNA have very similar LOH "ratios", but the background-subtracted meiotic LOH frequencies are almost 10-fold different: 1.9% versus 0.27%. In another example, SpNA/ScNA MSH2 and SpNA/ScNA msh2 Δ show a 2-fold difference in background-subtracted LOH ratios, but show similar LOH ratios (8 versus 5). Therefore, the conclusion that non-collinearity does not reduce LOH (line 131) is incorrect, and the impact of mismatch repair is modest, at best, using this assay (however, it appears to have a substantial impact when assayed by whole-genome sequencing). Many other conclusions may change, as well.

[R] We thank the reviewer for this insightful comment. We agree that this is an important aspect of the manuscript that needs to be further clarified and expanded. We believe that both metrics, the “LOH difference” and the “LOH ratio”, are useful to interpret the data and we have implemented both throughout the text. We have now defined the “LOH difference” ($R_{T6} - R_{T0}$) as well as the “LOH ratio” (R_{T6}/R_{T0}) and we have interpreted both when evaluating the overall impact of RTG. Both metrics are calculated from the LOH rates at T6 (R_{T6}) and T0 (R_{T0}), which we derived from the *URA3*-loss assay. Thus, they reflect the intrinsic constraints of the LOH rates (R_{T0} and R_{T6}).

The R_{T0} value is dominated by LOHs occurring during mitotic divisions (as proven by the high frequency of this event compared to chromosome loss and loss-of-function mutations). Instead, R_{T6} takes into account also the level of DSBs induction and the fraction of these DSBs that generate a LOH towards the subgenome bearing the *LYS2* allele. Assuming that no subgenome bias occurs, we can expect that, for each LOH event homogenising the *LYS2/URA3* marker in one direction, another one arose in the opposite direction - but it was not selected. Moreover, the *URA3*-loss assay excludes all the RTGs which do not recombine at the location of the marker but elsewhere in the genome. In other words, the assay is conservative and underestimates the impact of RTG.

The LOH difference and LOH ratio account for the mitotic contribution to LOH in two different ways. The LOH difference treats mitotic LOH events as a background signal that can be subtracted from the value observed in RTG, to estimate the absolute number of LOH events at the marker location that are due to RTG. The LOH ratio treats mitotic LOH events as background noise and provides a measure of the corresponding relative increase of local recombination upon RTG. We therefore agree with the editor and the other reviewers who were consulted on this issue that these metrics provide complementary information and that reporting both provides the most comprehensive overview of the data. For instance, the non-collinear ScMA/ScNA intraspecies hybrid shows a lower “LOH difference” compared to the homozygous ScNA/ScNA. This holds true also for the collinear intraspecies hybrids (e.g. ScWE/ScNA or ScWA/ScNA), including the ScWA/ScNA showing a higher LOH ratio compared to ScNA/ScNA (**Figure 2d**). Thus, the non-collinear ScMA/ScNA and the collinear ScWA/ScNA show similar trends, suggesting that the variability between ScNA/ScNA and ScMA/ScNA in the metric “LOH difference” is not due to the non-collinearity but it is more likely influenced by the level of sequence divergence between the subgenomes. Indeed, the sequence divergence across the intraspecies hybrids (in particular ScMA/ScNA and ScWA/ScNA) is similar (see **Figure 2b**).

Regarding the observation that *MSH2* deletion has a modest impact on improving RTG recombination in the SpNA/ScNA hybrid, we reported in the text (line 146-150) that this mutant had a slower meiotic progression compared to the wild-type SpNA/ScNA. RTG-induced recombination must be evaluated in terms of meiotic progression, i.e. the percentage of MI+MII cells measured by DAPI staining, and recombination efficiency. The former represents a first barrier to RTG-induced recombination. Thus, a comprehensive comparison also needs to take into account the differences in the speed and synchrony of meiotic progression, since it is an important factor affecting both the “LOH difference” and the “LOH ratio”. The slower meiotic progression of the *msh2Δ* mutant likely explains why the “LOH ratio” and the “LOH difference” increases seem modest, but the genome-wide impact is more substantial.

[A] We followed the reviewer’s suggestions and modified **Figures 2d** and **2e**, including the “subtracted meiotic LOH frequencies” (the “LOH difference”) to both plots. As suggested by the reviewer, we also added the percentage of MI+MII cells (meiotic progression) measured by DAPI staining at the bottom of panel **2d**.

Through the text we now use of the following terms:

- “LOH rate” when referring to the single time-points measures (e.g. R_{T0} is the LOH rate at T0 while R_{T6} is the LOH rate at T6) obtained from the *URA3*-loss assay.
- “LOH ratio” when referring to the R_{T6}/R_{T0} metric.
- “LOH difference” when referring to the $R_{T6} - R_{T0}$ metric.

Similar LOH ratios/differences between ScMA/ScNA and other intraspecies hybrids with a comparable level of sequence divergence (e.g. ScWA/ScNA) supports the claim that non-collinearity does not reduce RTG recombination, when measured by the *URA3*-loss assay. Indeed, the ScMA/ScNA had a comparable LOH difference to hybrids with similar meiotic induction and a much higher LOH difference compared with intraspecies hybrids having an inefficient meiotic induction (e.g. ScSA/ScNA, ScSA/ScWA).

Line 71—it remains unclear whether or not the Spo11 complex actually has topoisomerase activity. Perhaps “a homolog of DNA topoisomerase VI”.

[R&A] We thank the reviewer for pointing this out and we modified the introduction accordingly (lines 70-71).

Line 76—“bud like mitotic cells”

[R&A] We modified this point accordingly (line 75).

Line 94—perhaps it is worth explaining that the wild-type 4-5 h sample contains cells at different stages of meiosis, with different levels of meiotic DSBs, which would explain the heterogeneity of this population with regards to crossover numbers. This is stated later (line 104) but in a non-obvious way.

[R&A] We agree and modified the text accordingly (lines 95-96).

Line 137, Table S4. It was difficult to figure out which SpNA/ScNA hybrid was wild-type, *msh2Δ*, or *ndt80Δ* in Table S4. It would be useful to add this information.

[R&A] We added the information on these mutants to **Table S4** (second column).

Line 158, Table S9. It would be useful to include the extent of sequence divergence and number of departures from collinearity for the hybrid strains listed in Table S9.

[R&A] For all the crosses reported in **Table S9**, we added the sequence divergence and the number of departures from collinearity along with the total number of markers, the number of markers lying in collinear regions and the number of markers lying in non-collinear regions.

Line 170 and ff, Fig 3D and S4C. The association between breakpoints and DSB hotspots is presented in a way that is confusing and does not make the case clearly, and the section in the materials and methods is not very helpful. I am not sure how to improve this, but perhaps one could take the distribution of Spo11 oligo intensities in a region around each LOH breakpoint (suggest 2 kb on either side, but other distances might be tried) and compare it to the distribution of Spo11 oligo intensities genome-wide?

[R&A] We agree with this remark and clarified this point further by expanding the supplementary materials, supplementary tables, as well as the supplementary figures. We included a brief description of the statistical method used and the corresponding reference [Gel et al. 2016. Bioinformatics]. Following the reviewer’s suggestion, we also compared the distributions of Spo11 oligos intensity in regions overlapping the LOH breakpoints against the intensity in regions that do not overlap the LOH breakpoints. This comparison allowed for the calculation of the p-value for the one-sided Wilcoxon Rank-Sum test. The hotspots dataset provided by Mancera et al. (268 hotspots covering 416 kb) was not taken into account since it does not report intensity values. Only the hotspots data provided by Pan et al. were used (i.e. 3600 hotspots covering 908 kb).

We added different padding distances at both ends to expand the boundaries of the LOH breakpoints. We observed the following trend: the larger the padding, the lower the difference between the distributions of Spo11 oligos intensity in overlapping and non-overlapping regions. All the crosses showed that the intensity in overlapping regions was higher than the intensity in non-overlapping regions for padding values < 1000 bp. E.g., for the ScMA/ScNA hybrids we obtained the following genome-wide distributions:

ScMA/ScNA
 N (overlapping): 4210
 N (non-overlapping): 2987
 padding (bp): 5000

For each padding value, we obtained the following means, standard deviations, medians and p-values:

padding (bp)	relative intensity in overlapping regions				relative intensity in non-overlapping regions				p-value
	sample size	mean	standard deviation	median	sample size	mean	standard deviation	median	
0	1139	0.079	0.115	0.037	6058	0.043	0.064	0.022	1.1E-38
500	1795	0.069	0.103	0.033	5402	0.042	0.062	0.021	9.6E-35
1000	2266	0.065	0.098	0.031	4931	0.041	0.061	0.021	2.2E-32
2000	2910	0.061	0.093	0.029	4287	0.040	0.059	0.020	2.7E-30
5000	4210	0.054	0.085	0.026	2987	0.041	0.059	0.021	1.3E-13

The plots with no padding for the ScWE/ScNA and the ScMA/ScNA crosses were added in **Supplementary Figure 4d** while the tables (with all padding values) were included in **Table S11**.

Figure 2—The colors used in this figure are not so easy to differentiate from each other. Rather than numbering each class 1, 2, 3, 4 etc., authors might want to give each hybrid a unique number (i.e. homozygous parents 1,2,3,4; intraspecies hybrids 5,6,7, etc).

[R&A] We modified the numbers assigned and used a unique number for each hybrid or parental homozygous strain. We now use different shapes (triangles and dots) to diversify the homozygous parents from the intraspecies hybrids.

Table S13—please provide a legend to explain what the headings in “markers distribution per class” mean. In particular, what is meant by homozygous reciprocal, homozygous non-reciprocal, and homozygous 4:0? In general, the clarity of the supplementary tables would be considerably enhanced by the inclusion of explanatory legends that define each of the column headers.

[R] The terms refer to the patterns of markers segregation observed in the RTG mother/daughter pairs.

[A] We added a legend in the header of **Table S13** and we improved the naming of the columns whenever necessary. Regarding **Table S13**, the header provides a brief definition of the 4 classes of markers reported in the table:

- *heterozygous*: both mother and daughter cells are heterozygous.
- *homozygous reciprocal*: both the mother and the daughter cells are homozygous but with different alleles.
- *homozygous non-reciprocal*: one cell (e.g. the mother) is homozygous but the other one (e.g. the daughter) is heterozygous.
- *homozygous "4:0"*: both the mother and the daughter cells are homozygous, with the same allele.

Reviewer #2 (Remarks to the Author):

This is an impressive and methodologically thorough study providing interesting and novel content. I applaud the authors on this in-depth investigation of the return-to-growth (RTG) process in *Saccharomyces* yeasts, combining a large range of state-of-the-art approaches in yeast genetics, including laboratory hybrid crosses, genetic engineering, WGS, genomics analyses, high throughput phenomics, and QTL analysis. The authors show that normally sterile hybrid yeast strains can abort meiosis and return to mitotic growth, with the effect that a level of genomic recombination is maintained but reproductive species barriers are bypassed and fertility is (partly) restored in hybrids. The authors claim that this may represent a viable route to increased fitness and adaptation of yeast hybrids and other microbial eukaryotes that can reproduce both asexually and sexually.

While this study is certainly very interesting for a specialist audience, I think the authors may overstate the relevance of their findings a little, considering the broad range of readers of this journal. The return-to-growth mechanism is not likely to have the same wide-ranging effects on adaptive evolution in obligate sexual organisms with a more complex organisation and development. It would thus be preferable if the implications, especially with respect to hybrid speciation in the abstract and discussion, would be given some more qualifying conditions.

[R&A] We agree that an RTG-like mechanism is not likely to have an impact on complex obligate sexual organisms and revised the text accordingly. We modified and included in the abstract and discussion more qualifying conditions whenever necessary (e.g. lines 313-315). As suggested by reviewer 3, we have revised the aspects related to overcoming hybrid sterility and we have modified the title of the manuscript accordingly.

The other point I think the authors are overstating is the recombination rate increase, or the size of the RTG effect on LOH. It seems to me that this effect is highly cross dependent (which should be tested) and may only confer a slight increase compared to mitosis (Figure 2D). I don't think the effect is directly comparable to outcomes of sexual recombination, which includes segregation and causes new combinations of alleles within genomes.

[R] We improved the description of the metrics we used to assess the recombination rates (as described in the reply to Reviewer 1). We also provided a novel metric, the "LOH difference", to quantify the magnitude of RTG-induced LOH, as suggested by reviewer 1. The definitions of the metrics are provided in the supplementary material.

Here are some more specific comments:

1. Have you tested whether recombination rates were correlated to sequence-wide genetic distance between parental strains in your data?

[R] We used a “simple linear regression model” approach to fit the recombination rates data (measured as both LOH difference and LOH ratio) against the sequence-wide genetic distance data but we did not find significant evidence to support the regression (correlation “LOH ratio”-“sequence divergence”: $r=-0.36$, $p=0.23$, Correlation “LOH difference”-“sequence divergence”: $r = -0.5$, $p = 0.08$; r is the Pearson correlation coefficient). We extended the correlation analysis between the LOH ratio and the meiotic progression (“MI+MII cells”) we had previously performed also to the LOH difference and the meiotic progression. The latter confirmed strong correlation.

We also used a multiple linear model taking into account two explanatory variables, i.e. the meiotic progression (“MI+MII cells”) and the sequence divergence (“Sequence Divergence”), and one explained variable. Using the LOH ratio as the explained variable, the sequence divergence did not show statistical evidence to support the regression. Thus, the multiple linear regression model could be reduced to the simple model that we used in **Figure 2d**.

The values reported in the following table are: the estimated coefficient (“Estimate”), its standard error (“Std. Error”), the t-statistic (“t value”) and the corresponding two-sided p-value (“Pr(>|t|)”).

#Multiple linear regression using the relative LOH ratio
Coefficients:

	Estimate	Std. Error	t value	Pr(> t)
Intercept	5.246e-02	2.909e-01	0.180	0.860
MI+MII cells	6.445e-02	9.019e-03	7.146	3.12e-05 ***
Sequence Divergence	-3.983e-07	2.876e-07	-1.385	0.196

Signif. codes: 0 ‘***’ 0.001 ‘**’ 0.01 ‘*’ 0.05 ‘.’ 0.1 ‘ ’ 1

Residual standard error: 0.5225 on 10 degrees of freedom
Multiple r-squared: 0.8504, Adjusted r-squared: 0.8204
F-statistic: 28.42 on 2 and 10 DF, p-value: 7.501e-05

Sequence divergence is a marginally significant explanatory variable for the LOH difference. However, the major explanatory variable in the model remains the meiotic progression.

#Multiple linear regression using the LOH difference
Coefficients:

	Estimate	Std. Error	t value	Pr(> t)
Intercept	1.419e-02	3.618e-02	0.392	0.703148
MI+MII cells	5.152e-03	1.122e-03	4.593	0.000991 ***

Sequence Divergence	-9.190e-08	3.578e-08	-2.568	0.027967 *
------------	-----------	--------	------------

Signif. codes: 0 ‘***’ 0.001 ‘**’ 0.01 ‘*’ 0.05 ‘.’ 0.1 ‘ ’ 1

Residual standard error: 0.065 on 10 degrees of freedom

Multiple r-squared: 0.7595, Adjusted r-squared: 0.7114

F-statistic: 15.79 on 2 and 10 DF, p-value: 0.0008047

[A] We added a paragraph in the supplementary text (Testing for “recombination rates”-“meiotic progression” correlations and “recombination rates”-“sequence divergence” correlations) that describes the results reported above.

2. Line 124: The text mentions these cross abbreviations here without explanation. It took me a while, and a look at Figure 2, to realize that the ScMa/ScNA hybrid cross involves the highly non-collinear Malaysian strain. It would be good to mention this interesting detail more explicitly in the main text.

[R&A] We agree with the reviewer and modified the text accordingly (lines 134-136).

3. Line 137-142: I see no details of the *msh2*-deleted strain in Figure 2D or 2E at all, although the text refers to it.

[R&A] We modified the text, which had the wrong references to **Figure 2d** and **2e**.

4. Figure 1D. Is there a better way to display the mean/median and variance between clones to compare between these two plots? The Figure is not very convincing with respect to the point you are trying to make in the text (line 103 “the distribution was more homogeneous in the *ndt80Δ* population compared to the wild-type population.”)

[R&A] We agree and added a boxplot showing the fraction of markers in LOH (y-axis) for each dataset (WT and *ndt80Δ*) to **Figure 1d**. Each point represents a single sample of a mother/daughter RTG pair.

5. Line 162: “This underscores that genome-wide RTG recombination is not hampered by extensive non-collinearity between subgenomes, in line with that the sterility caused missegregation of the non-collinear chromosomes rather than a lack of recombination.” I don’t understand the second half of this statement. In my understanding, anti-recombination causes missegregation, which in turn causes sterility (i.e. highly aneuploid, unviable gametes) of F1 hybrids. So in my mind, the causality is the other way around than you imply here.

[R&A] We modified the main text accordingly (lines 172-175).

6. Line 165: “Moreover, neither parental subgenome was favoured over the other in terms of the created homozygosity”. I am not suggesting you adding more work here but I am really intrigued why the two subgenomes are so well balanced in your hybrids. Why are they not more affected by the selective environment and/or epistasis as seen by others before (Smukowski Heil et al, MBE, 2017; Zhang & Bendixsen et al MBE 2019)? You are alluding to it briefly in line 276 (“LOH regions might be selected by adaptation under specific selective regimes but may also be constrained by incompatibility.”) Would you expect an interaction between

homozygosity and the environment here, e.g. do you expect to find larger homozygosity of the *S. cerevisiae* subgenome in media where the *S. cerevisiae* parent has higher fitness than *S. paradoxus*?

[R] We thank the reviewer for his/her interest in this aspect of our work.

[A] We added a sentence in the discussion where we explain why, according to our experimental design, we observed no parental bias in LOH formation in contrast with what has been observed in adaptive evolution experiments (lines 297-299). Briefly, in RTG selective pressure is minimised as samples are grown for a very limited number of generations in non-selective rich media.

7. It would be good to see these results discussed in a more quantitative genetics context, e.g. considering the fact that many high fitness alleles have been found to be recessive and are only expressed in the homozygous state (Zörgö et al. MBE 2012; Bernardes et al. JEB, 2017). Do the authors consider their results to be important with respect to the role of dominance, overdominance or underdominance in hybrid fitness (e.g. as seen in Laiba et al. Genome, 2016)?

[R&A] We have now reported the analysis on the cases of best and worst parent heterosis (**Table S14**) (lines 241-248). The comparison of these results and the suggested references has been included in the discussion (lines 301-306). The prevalence of the worst parent heterosis observed in the RTG is consistent with our previous study using Phased Outbred Lines (POLs, described in Hallin et al 2016. Nature Communications), where large regions of homozygosity can unmask recessive loss-of-function variants. In contrast, the whole genome heterozygosity produced in F1 hybrids contributes towards mid- and best- parent heterosis. Given the large size of linkage blocks, we are unable to fine-map and characterise these loci, as we did in Hallin et al. However, we were able to connect a large fraction of the worst parent heterosis to the large CNVs observed in the ScMA/ScNA RTGs.

8. Legend to Figure 2, line 456: "...panels B, E and E".

[R&A] We fixed the wrong letter of the panel in the legend.

9. Update Bozdag et al reference (now published in Current Biology <https://doi.org/10.1016/j.cub.2020.12.038>)

[R&A] We kindly ask to maintain the reference to the original Biorxiv version of this work because the Current Biology version has been drastically shortened and no longer contains the genetic incompatibilities, which is what we refer to.

Reviewer #3 (Remarks to the Author):

This is an interesting and thorough paper showing that inducing meiosis by starving yeast cells, but then restoring nutrients before meiosis progresses, causes homologous recombination between the diploid chromosomes resulting in loss of heterozygosity (LOH, i.e. gene conversion). Normal mitotic recombination also produces such LOH events during normal diploid growth, but the return to growth method (RTG) greatly increases the rate, as shown in Figure 2D. This is a clever idea, and a potentially useful method for mapping phenotypes in sterile heterozygotes, and the authors prove the principle with a couple of nice experiments. Overall, the work is interesting and useful and clearly presented.

However, I have one major criticism which I think prevents it from being published in Nature Communications. The authors present the RTG technique as though it overcomes hybrid sterility, for example in the title, but it

doesn't. What is usually meant by hybrid sterility is the inability of a hybrid to complete its sexual cycle successfully. Sex comprises the production of gametes (meiosis) followed by the fusion of gametes (syngamy). Sex has two genetic effects, segregation and recombination, which together can increase or decrease the genetic diversity within the diploid genomes of individuals of the next generation, and within the populations they comprise. Hybrid sterility in yeast usually means the inability of hybrid diploids to produce viable haploid gametes by meiosis, and this is what the authors claim to have overcome.

[R&A] We thank the reviewer for the remark. We have split it into two different comments which we have addressed separately.

1. The authors consider three forms of sterility, e.g. in the introduction and in Figure 1. The first is the loss of function of meiotic genes, so that meiosis doesn't progress (this isn't strictly "hybrid sterility" since it is likely to affect non-hybrids too, unless the authors are suggesting that hybrids are more susceptible because of incompatibility between meiosis gene allele from different species which is plausible, but not made explicit).

[R] We agree with the reviewer. Indeed, when addressing the first form of sterility (loss-of-function mutations in meiotic genes) we refer to the *S. cerevisiae* populations described in De Chiara, M. et al. Domestication reprogrammed the budding yeast life cycle. [biorxiv.org. doi:10.1101/2020.02.08.939314](https://doi.org/10.1101/2020.02.08.939314). These populations include intraspecies *S. cerevisiae* hybrids with a broad spectrum of heterozygosity.

[A] We modified the corresponding paragraph of the introduction: "Relaxed selection on sexual reproduction in domesticated populations of *S. cerevisiae*, including intraspecies hybrids, has led to the accumulation of loss-of-function mutations in genes involved in gametogenesis, i.e. "sporulation" in yeast biology...".

2. The authors show that in each of these types of sterile hybrids, RTG increases recombination between chromosomes, so that diploids with LOH are produced. However, this is not the equivalent of successfully overcoming hybrid sterility. Sex produces gametes which can outcross, increasing genetic diversity and producing novel combinations of alleles. RTG doesn't produce gametes, and LOH only results in the loss of genetic diversity, without producing novel combinations. RTG does some of the things that sex does, but it's not, in my opinion, "overcoming hybrid sterility".

[R] We agree with the reviewer and modified the title accordingly. RTG offers a direct route to "bypass", rather than "overcome", one of the issues related to hybrids' sterility. The generation of wide-spread LOH tracts may allow to "overcome" hybrid sterility by producing islands of sequence homology which can mediate successful meiotic recombination and lead to correct chromosomal segregation [D'Angiolo *et al.* 2020]. We partially showed such effect in the ScMA/ScNA hybrid in **Figure 6c-d**. However, we were unable to produce such extensive LOH patterns in the interspecies Sc/Sp hybrids to overcome sterility. Nevertheless, RTG directly enables recombination, thus promoting the evolution of sterile hybrids, and the LOHs generated may result in novel haplotype combinations forcing chimeric protein interactions and producing chimeric genes at LOH breakpoint sites.

[A] We modified the title to: "Aborting meiosis allows recombination in sterile diploid hybrids". We also modified the main text to clarify that RTG does provide an opportunity to recombine although it does not overcome the sterility of the hybrids. However, it can lessen the sterility by providing regions of homology that mediate homologous recombination and increase gamete viability, as shown in **Supplementary Figure 6a** and lines 192-198. In the discussion, we suggested that RTG can have a role in unmasking deleterious recessive alleles thus exposing them to selection.

REVIEWERS' COMMENTS

Reviewer #1 (Remarks to the Author):

The authors have addressed my previous concerns in the current revision. The following comments are more aimed at clarification of the text.

1. In many places in the manuscript, the term "evolved" is used to refer to cultures that have been through a single cycle of RTG. This is a bit confusing, since in the experimental evolution context "evolved" usually means having been through multiple cycles of selection and/or bottlenecks. Perhaps a statement at the onset defining clearly what is meant by "evolved" would help.
2. page 4 line 106--prophase
3. page 5 line 156. Shouldn't this just be Fig. 2d? I don't think that the ndt80 data are included in panel e.
4. page 6, line 166 and following. In RTG of ScMA/ScNA, is the frequency of recombination the same on the chromosomes that are rearranged as on chromosomes that are not rearranged?
5. page 7, line 230. The phrase "even a single RTG cycle" leaves the impression that multiple RTG cycles were also examined, which I don't think was the case.

Reviewer #2 (Remarks to the Author):

It looks like Reviewer 3 and I largely agreed on the main criticism that RTG and the two outcomes of sexual recombination, segregation and recombinations, are not directly comparable. I do think the authors have now considered this in their revision and have made according changes.

REVIEWERS' COMMENTS – 2nd round

Reviewer #1 (Remarks to the Author):

The authors have addressed my previous concerns in the current revision. The following comments are more aimed at clarification of the text.

1. In many places in the manuscript, the term "evolved" is used to refer to cultures that have been through a single cycle of RTG. This is a bit confusing, since in the experimental evolution context "evolved" usually means having been through multiple cycles of selection and/or bottlenecks. Perhaps a statement at the onset defining clearly what is meant by "evolved" would help.

[R&A] We agree and edited the text accordingly.

2. page 4 line 106—prophase

[R&A] Edited

3. page 5 line 156. Shouldn't this just be Fig. 2d? I don't think that the ndt80 data are included in panel e.

[R&A] Edited

4. page 6, line 166 and following. In RTG of ScMA/ScNA, is the frequency of recombination the same on the chromosomes that are rearranged as on chromosomes that are not rearranged?

[R&A] This has been clarified in the text and in the supplementary notes.

5. page 7, line 230. The phrase "even a single RTG cycle" leaves the impression that multiple RTG cycles were also examined, which I don't think was the case.

[R&A] We removed "even" from the sentence.

Reviewer #2 (Remarks to the Author):

It looks like Reviewer 3 and I largely agreed on the main criticism that RTG and the two outcomes of sexual recombination, segregation and recombinations, are not directly comparable. I do think the authors have now considered this in their revision and have made according changes.